# Leveraging Submodule Linearity Enhances Task Arithmetic Performance in LLMs

**Rui Dai** [1]  **Sile Hu** [2]  **Xu Shen** [* 2]  **Yonggang Zhang** [3]  **Xinmei Tian** [* 1]  **Jieping Ye** [2]

[1]National Engineering Laboratory for Brain-Inspired Intelligence Technology and Application, University of Science and Technology of China
[2]Independent Researcher
[3]Hong Kong Baptist University
*shenxuustc@gmail.com, xinmei@ustc.edu.cn

## Abstract

Task arithmetic is a straightforward yet highly effective strategy for model merging, enabling the resultant model to exhibit multi-task capabilities. Recent research indicates that models demonstrating linearity enhance the performance of task arithmetic. In contrast to existing methods that rely on the global linearization of the model, we argue that this linearity already exists within the model's submodules. In particular, we present a statistical analysis and show that submodules (e.g., layers, self-attentions, and MLPs) exhibit significantly higher linearity than the overall model. Based on these findings, we propose an innovative model merging strategy that independently merges these submodules. Especially, we derive a closed-form solution for optimal merging weights grounded in the linear properties of these submodules. Experimental results demonstrate that our method consistently outperforms the standard task arithmetic approach and other established baselines across different model scales and various tasks. This result highlights the benefits of leveraging the linearity of submodules and provides a new perspective for exploring solutions for effective and practical multi-task model merging.

## 1 Introduction

In recent years, the growing scale of large language models has significantly increased the demand for data and training costs in fine-tuning multi-task models. To integrate the capabilities of various existing single-task models into a unified model, various model merging techniques have been developed (Yu et al., 2024; Jin et al., 2023; Matena & Raffel, 2022; Yadav et al., 2023). Task arithmetic (Ilharco et al., 2023) is one of the simplest and most efficient model merging strategies. It involves a straightforward weighted combination of the weights from single-task models, allowing the final merged model to exhibit multi-task capabilities without the need for additional training or data.

Recent studies (Ortiz-Jiménez et al., 2023; Zhou et al., 2024) have revealed a connection between the linearity of fine-tuned models and the effectiveness of the weighting operations in task arithmetic. Here, *linearity* specifically refers to the linear relationship between the differences in model weights and the differences in output features caused by fine-tuning[1] (Ortiz-Jiménez et al., 2023; Zhou et al., 2024), which is very different from the Traditional Linear Properties[2]. This linearity intuitively aligns with the linear weighting operations in task arithmetic (Ilharco et al., 2023). Research (Ortiz-Jiménez et al., 2023; Tang et al., 2024; Jin et al., 2024; Liu et al., 2024) has shown that models exhibiting linearity retain their individual task performance better when merged using task arithmetic, leading to superior multi-task models. However, models produced through conventional fine-tuning often lack this ideal linear property (Ortiz-Jiménez et al., 2023). To address this issue, existing methods (Ortiz-Jiménez et al., 2023; Tang et al., 2024; Jin et al., 2024; Liu et al., 2024) have

---

*Corresponding authors. Code: https://github.com/deep-analysis-research/SLTA.

[1]$f(x; \theta_0 + \alpha\tau) \approx f(x; \theta_0) + \alpha\Delta f(x; \theta_0 + \tau)$ where $\tau = \theta - \theta_0$ is the weight difference caused by fine-tuning and $\Delta f(x; \theta_0 + \tau) = f(x; \theta_0 + \tau) - f(x; \theta_0)$ is the corresponding output feature difference.

[2]These two properties are easily confused. More discussions see Appendix A.5.1.

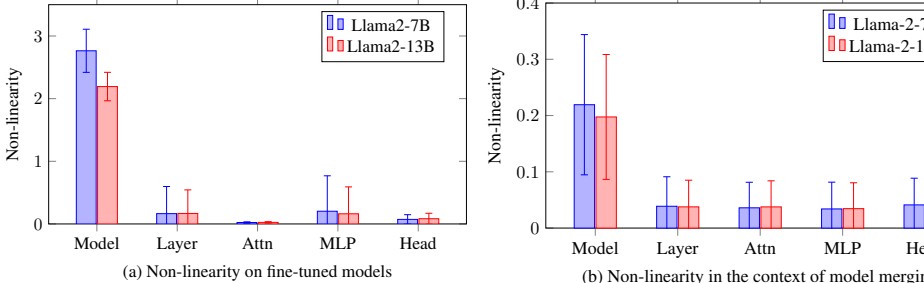

Figure 1: (a) Comparison of non-linearity in full models and submodules for fine-tuned models, measured using Non-linearity Score defined in Definition 2. A lower value indicates better linearity, based on data from three fine-tuned models detailed in Section 4.1. (b) Comparison of non-linearity in full models and submodules within the context of model merging, assessed via Eq.8 (Zhou et al., 2024). A lower value indicates better linearity, based on data from three merging combinations outlined in the caption of Figure 3. Error bars represent standard deviation.

explored adjustments to the fine-tuning paradigm, retraining the model using original training data to enhance the linearity and the final merging outcomes.

In contrast to existing methods that rely on global linearization of the model, which impose requirements for retraining with the original data, we argue that this linearity has already existed within the model's submodules. Intuitively, the most straightforward approach to seek linearity from a complex nonlinear function is to decompose it into simpler ones. Following this idea, we divide the model into multiple shallower submodules (e.g., layers, self-attentions, MLPs) and discover that these submodules exhibit a level of linearity that significantly surpasses that of the overall model. Figure 1 compares the linearity characteristics of full models and their constituent modules. In panel (a), the non-linearity of full models is significantly higher than that of the decomposed submodules, which exhibit scores close to zero, indicating their strong linearity. In panel (b), we further evaluate the linear properties in the context of model merging[3] using the metric introduced by Zhou et al. (2024). The results reveal a distinct gap in non-linearity between full models and submodules, suggesting that the latter can demonstrate good linearity even when the full models do not.

Inspired by these observations, we propose an innovative training-free model merging strategy: 1) Decompose the models into submodules that exhibit linear characteristics. 2) Compute the optimal merging weights for each submodule by leveraging their linear properties. 3) Apply these derived weights to facilitate the merging of the models. To obtain the optimal merging weights under the linearity assumption of the modules, we derived a closed-form solution by minimizing the distance between the output features of the merged module and those of the corresponding module across fine-tuned tasks (Jin et al., 2023). Notably, this computational process necessitates only a minimal amount of data (30 samples per task) and a limited number of model inference iterations.

To validate the effectiveness of our proposed method, we conducted merging experiments across models fine-tuned from Llama-2-7B and Llama-2-13B (Touvron et al., 2023) for three distinct tasks: Math, Coding, and Translate. The experimental results demonstrate that our method significantly outperforms various baseline techniques, including standard arithmetic approaches, across different model scales and diverse tasks, particularly with respect to the decomposition level of layers and attn/MLPs. These findings provide strong evidence that leveraging the linearity inherent in submodules holds substantial potential for enhancing model merging performance with task arithmetic.

## 2 ANALYZING THE LINEARITY OF MODEL AND MODULES

In this section, we first introduce some definitions regarding linearity (Ortiz-Jiménez et al., 2023) and task arithmetic(Ilharco et al., 2023). We then present a score that can quantitatively compare the

---

[3] $f(x; \theta_0 + \sum_{t=1}^{T} \alpha_t \tau_t) \approx f(x; \theta_0) + \sum_{t=1}^{T} \alpha_t \Delta f(x; \theta_0 + \tau_t)$, where $\tau_t = \theta_t - \theta_0$ is the weight difference caused by fine-tuning in the $t$th model and $\Delta f(x; \theta_0 + \tau_t) = f(x; \theta_0 + \tau_t) - f(x; \theta_0)$ is the corresponding output feature difference in the $t$th model.

linearity of models and modules. Subsequently, following the insight outlined in the introduction, we divide the model into submodules to examine linearity, discovering that the linearity of the module surpasses that of the model. Finally, we further validate the linearity when merging multiple modules, laying a foundation for the methodologies proposed in the next section.

## 2.1 PRELIMINARY

**Notation.** Let $f : X \times \Theta \to Y$ be a neural network taking inputs $x \in X$ and parameterized by a set of weights $\theta \in \Theta$. Consider $T$ tasks, where $D_t \subseteq X$ is a dataset used to fine-tune the models starting from the pre-trained weights $\theta_0$ and obtain the fine-tuned weights $\theta_t$.

**Task Arithmetic.** (Ilharco et al., 2023) Task arithmetic is a simple, efficient, and widely used model merging strategy. It assigns multi-task performance to the merged model through a simple weighted combination of the weights of models, without the need for additional training or data support.

*Task vector* for task $t$ is defined as the difference between the fine-tuned and the pre-trained weights, namely, $\tau_t = \theta_t - \theta_0$. The approach of task arithmetic involves a weighted combination of the task vectors, which is then added to the pre-trained model $\theta_0$. This can be expressed as

$$\theta_{\text{merge}} = \theta_0 + \sum_{t=1}^{T} \alpha_t \tau_t, \tag{1}$$

where $\alpha_t$ represents the weight corresponding to $\tau_t$.

Here, *linearity* specifically refers to the linear relationship between the differences in model weights and the differences in output features caused by fine-tuning

**Definition 1** *(Linearity). Let $\theta$ be a fine-tuned model and $\theta_0$ be its corresponding pre-trained model. If we assert that $\theta$ exhibits linearity, the differences in model weights caused by fine-tuning are linearly related to the differences in output features caused by fine-tuning for any $x \in X$, i.e.*

$$f(x; \theta_0 + \alpha\tau) \approx f(x; \theta_0) + \alpha\Delta f(x; \theta_0 + \tau) \tag{2}$$

*where $\tau = \theta - \theta_0$ is the differences in model weights before and after fine-tuning, and $\Delta f(x; \theta_0 + \tau) = f(x; \theta_0 + \tau) - f(x; \theta_0)$ is the differences in model output features before and after fine-tuning. It is very different from the Traditional Linear Properties.*[4]

Research (Ortiz-Jiménez et al., 2023; Jin et al., 2024) has shown that models demonstrating linearity retain higher single-task performance when merged via task arithmetic, leading to enhanced multi-task models. However, models developed through conventional fine-tuning methods typically lack this desired linearity (Ortiz-Jiménez et al., 2023).

**Proposed Linearity Metric.** The interpolation model $\theta_{\text{inter}}$ between the model $\theta$ and the pre-trained model $\theta_0$ lies on the line connecting $\theta$ and $\theta_0$ in the parameter space. If the model $\theta$ satisfies linearity in Definition 1, then its output features $f(\theta_{\text{inter}}, x)$ should also lie on the line formed by the output features $f(\theta, x)$ and $f(\theta_0, x)$. Moreover, the distance between $f(\theta_{\text{inter}}, x)$ and $f(\theta_0, x)$ should be proportional to the interpolation weight. Based on this idea, to measure and compare the linearity of models quantitively, we propose the *Non-linearity Score*.

**Definition 2** *(Non-linearity Score). For a fine-tuned model $\theta$ and its corresponding pre-trained model $\theta_0$, if $\theta$ satisfies linearity in Definition 1 under any distance metric $\mathcal{D}(\boldsymbol{a}, \boldsymbol{b}) = \|\boldsymbol{a} - \boldsymbol{b}\|$ in the feature space, where $\|\cdot\|$ denotes the norm operation, then for any $\alpha, \beta \in [0, 1]$, the following approximation holds:*

$$\mathcal{D}(f(x; \theta_0 + \alpha\tau), f(x; \theta_0 + \beta\tau)) \approx |\alpha - \beta| \, \mathcal{D}(f(x; \theta), f(x; \theta_0)) \tag{3}$$

*where $\tau = \theta - \theta_0$.*

---

[4]The preliminary investigation into the relationship between task arithmetic and linearity commenced with the linearity conclusion derived from the research on the Neural Tangent Kernel (NTK)(Jacot et al., 2018; Chizat et al., 2019). It was established that the output of a neural network can be approximated using a first-order Taylor expansion: $f(x; \theta_0 + \alpha\tau) \approx f(x; \theta_0) + \alpha\tau^{\top}\nabla_\theta f(x; \theta_0)$, which presents a more rigorous conclusion.

*When we replace the approximation in Eq.3 with equality, it indicates a state of perfect linearity. Therefore, we define the Non-linearity Score of model θ as the measure of the deviation of model θ from this ideal linear condition:*

$$\text{Non-linearity Score} = \int_0^1 \int_0^1 \left( \frac{\mathcal{D}(f(x;\theta_0 + \alpha\tau), f(x;\theta_0 + \beta\tau))}{\mathcal{D}(f(x;\theta), f(x;\theta_0))} - |\alpha - \beta| \right)^2 \mathrm{d}\alpha\mathrm{d}\beta \quad (4)$$

We calculated the Non-linearity Score [5] for models fine-tuned on various tasks and presented the results in Figure 2. Our observations indicate that nearly all models exhibit a lack of satisfactory linearity at the model level. This finding aligns with conclusions drawn in other research Ortiz-Jiménez et al. (2023) employing different methodologies.

## 2.2 ANALYZING THE LINEARITY OF SINGLE FINE-TUNED MODELS

Following the idea of seeking linearity by decomposition, we divide the entire model according to its architecture into multiple levels, including layers, self-attention, MLP, and head levels. Analogous to Definition 1, the linearity at the module level can be expressed as follows:

**Property 1** *(Linearity of Module). Let $\theta^i$ be a module weight from the fine-tuned model $\theta$ and $\theta_0^i$ be its corresponding module's weight from the pre-trained model. If we assert that $\theta^i$ exhibits linearity, the differences in module weights caused by fine-tuning are linearly related to the differences in this module's output features caused by fine-tuning for any $x \in H^i$, i.e.*

$$f(x;\theta_0^i + \alpha\tau^i) \approx f(x;\theta_0^i) + \alpha\Delta f(x;\theta_0^i + \tau^i) \tag{5}$$

*where $\tau^i = \theta^i - \theta_0^i$ is the differences in module weight before and after fine-tuning, and $\Delta f(x;\theta_0^i + \tau^i) = f(x;\theta_0^i + \tau^i) - f(x;\theta_0^i)$ is the differences in module output features before and after fine-tuning with same input $x \in H^i$, $H^i$ corresponds to the input feature space of module i.[6]*

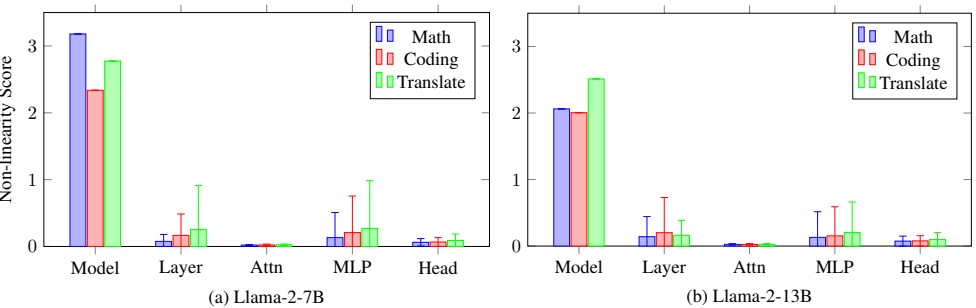

Figure 2: Comparison of Non-linearity Score in full models and submodules for models fine-tuned from tasks of Math, Coding, and Translate in Llama-2-7B and Llama-2-13B. A lower value indicates better linearity. Error bars represent standard deviation across different modules.

**Linearity in Models and Modules.** We can employ the same Non-Linearity Score defined in 2 to assess the degree of satisfaction concerning this property for multiple fine-tuned models and different sets of corresponding submodules. In Figures 2(a) and (b), we compare the average linearity at both the module level and model level across models fine-tuned on three distinct tasks, on two different scales of Llama-2 backbones. Several observations can be derived from these figures: 1) All submodules exhibit a degree of linearity that significantly *exceeds* that of the overall model. 2) The Non-Linearity scores for all submodules are *near zero*, suggesting an adequate approximation as indicated in Eq.5. 3) The pronounced linearity disparity between the model level and the module level remains *consistent* across all three fine-tuned tasks and both scales of the model.

---

[5]For practical reasons, in this paper, we employ a discrete approximation of the metric, which we continue to refer to as the Non-linearity Score for simplicity. Further details can be found in Appendix A.2.

[6]We make a slight abuse of notation $f$ by omitting the subscripts that distinguish between different architectures for the sake of simplicity.

In conclusion, these observations provide compelling evidence that *the modules can exhibit linearity even when the models do not*. This suggests a considerable opportunity to leverage the linear properties at the submodule level when integrating fine-tuned models with task arithmetic, even if the overall model does not exhibit the same degree of linearity.

## 2.3   ANALYZING THE LINEARITY OF MERGING MULTIPLE FINE-TUNED MODELS

Property 1 pertains to the linearity of modules within a single fine-tuned model. However, in the context of model merging, multiple fine-tuned models are involved. Consequently, we extend Property 1 to encompass multiple models:

**Property 2** *(Linearity of Merged Modules). Let $\theta^i$ be a module weight from the fine-tuned model $\theta$ and $\theta_0^i$ be its corresponding module's weight from the pre-trained model. If we assert that $\theta^i$ exhibits linearity of merging, we imply that the differences in output features originating from different models can be linearly separated, i.e., for any $x \in H^i$,*

$$f(x; \theta_0^i + \sum_{t=1}^{T} \alpha_t^i \tau_t^i) \approx f(x; \theta_0^i) + \sum_{t=1}^{T} \alpha_t^i \Delta f(x; \theta_0^i + \tau_t^i) \qquad (6)$$

*where $\tau_t^i = \theta_t^i - \theta_0^i$ is the differences in module $i$ weight from model $t$ and $\Delta f(x; \theta_0^i + \tau_t^i) = f(x; \theta_0^i + \tau_t^i) - f(x; \theta_0^i)$ is the differences in module output features before and after fine-tuning.*

**Assesing Linearity in the Context of Model Merging.** Property 2 is only feasible in the presence of Property 1 for all corresponding models; however, its occurrence is not guaranteed. The combination of multiple models fine-tuned for distinct tasks may introduce additional non-linearity due to the approximations inherent in these equations. To ascertain whether the previously mentioned observations regarding linearity remain applicable within the context of model merging, we adopted the methodology delineated by Zhou et al. (2024), which evaluates linearity using a feature space metric. For clarity and convenience, we refer to this metric as Projection Distance in this paper.

Specifically, we firstly calculate the cosine similarity between two sets of feature differences: the differences between the output features of the merged module (derived from multiple fine-tuned models) and those of the pre-trained module, denoted as $\Delta f(x; \theta_0^i + \sum_{t=1}^{T} \alpha_t^i \tau_t^i) = f(x; \theta_0^i + \sum_{t=1}^{T} \alpha_t^i \tau_t^i) - f(x; \theta_0^i)$, as well as the corresponding weighted combination of the differences in output features from each fine-tuned model, represented as $\sum_{t=1}^{T} \alpha_t^i \Delta f(x; \theta_0^i + \tau_t^i) = \sum_{t=1}^{T} \alpha_t(f(x; \theta_0^i + \tau_t^i) - f(x; \theta_0^i))$. This similarity is denoted by the following equation

$$\text{cosine}_{merge}(x; \alpha_{1\ldots T}^i) = \cos\left(\Delta f(x; \theta_0^i + \sum_{t=1}^{T} \alpha_t^i \tau_t^i), \sum_{t=1}^{T} \alpha_t^i \Delta f(x; \theta_0^i + \tau_t^i)\right) \qquad (7)$$

where $\cos(\boldsymbol{a}, \boldsymbol{b}) = \boldsymbol{a} \cdot \boldsymbol{b}/(\|\boldsymbol{a}\|\|\boldsymbol{b}\|)$ is the Cosine Similarity. As follow, the Projection Distance $\text{proj}_{merge}^{\mathbb{E}}(\alpha_{1\ldots T}^i)$ between these two sets of feature deltas is represented as

$$\text{proj}_{merge}^{\mathbb{E}}(\alpha_{1\ldots T}^i) = |1 - \mathbb{E}_{x \in H^i} \frac{\|\Delta f(x; \theta_0^i + \sum_{t=1}^{T} \alpha_t^i \tau_t^i)\|\text{cosine}_{merge}(x; \alpha_{1\ldots T}^i)}{\|\sum_{t=1}^{T} \alpha_t^i \Delta f(x; \theta_0^i + \tau_t^i))\|}| \qquad (8)$$

where $\mathbb{E}$ represents expectation and $H^i$ represents the input feature set corresponding to module $i$, with further details provided in Appendix A.3.3.

**Results of Projection Distance**[7]. In Figure 3 (a) and (b), we present the projection distances resulting from the merging of two and three models. Both configurations reveal a trend that submodules exhibit superior linearity compared to the model level, consistent with the observations made for single models presented in Figure 2. It is noteworthy that the projection distance for merging three models is significantly lower than that observed for the merging of two models across layers, attentions, and MLPs. This suggests that there exists an even more favorable condition for linear merging

---

[7]Follow Zhou et al. (2024), we also analyze the metric of cosine similarity, please refer to Appendix A.1.1 for more results and analysis.

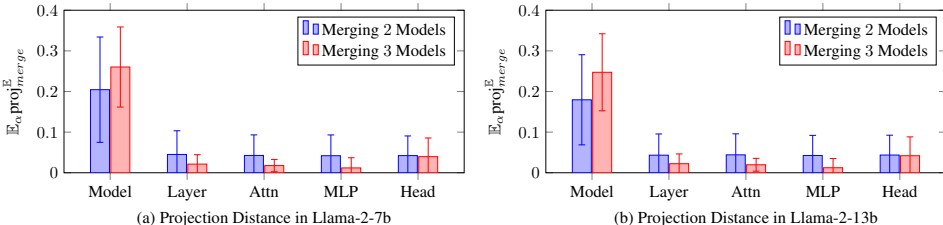

(a) Projection Distance in Llama-2-7b    (b) Projection Distance in Llama-2-13b

Figure 3: Comparison of Projection Distances in full models and submodules within the context of model merging. A lower value indicates better linearity. For each backbone, we computed the results across all possible combinations of three fine-tuned models, alongside the corresponding merging weights $\alpha_1$ and $\alpha_2$ (and $\alpha_3$). In the case of merging two models, there are three combinations of fine-tuned models, yielding 25 possible configurations for merging weights $\alpha_t \in [0.2, 0.4, 0.6, 0.8, 1]$ where $t \in [1, 2]$. For merging three models, there's only one combination of fine-tuned models, resulting in 27 configurations for merging weights $\alpha_t \in [0.3, 0.5, 0.7]$ with $t \in [1, 2, 3]$. Error bars represent standard deviations.

when three models are considered compared to when only two models are merged, particularly when analyzed at these granular levels.

In summary, these findings *validate the applicability of Property 2 within submodules*, demonstrating its relevance to the merging of both two and three models. As illustrated in Section 3, this property facilitates a closed-form solution for determining the optimal merging weights, thereby enhancing the task arithmetic performance while minimizing associated costs.

## 3  MERGING MODULES WITH LINEARITY

The selection of merging weights is a pivotal factor influencing model merging performance in task arithmetic. In (Ilharco et al., 2023), the authors propose determining this parameter through ablation while maintaining equal weights across different models to constrain the search space within a manageable range. Although this method demonstrates a commendable balance between cost and efficiency at the model level, it becomes impractical when merging models at the module level. In such cases, the search space expands exponentially relative to the number of modules; for instance, ablating $n$ candidate weights results in a search space of $n^m$ for $m$ modules. Fortunately, by leveraging Property 2, we can derive a closed-form solution for the optimal merging weights, thereby circumventing the cumbersome ablation process and expanding the solution space by relaxing the constraint of equal weights. Here we first introduce a commonly used objective in model merging (Jin et al., 2023; Ortiz-Jiménez et al., 2023), which is also regarded as the objective of task arithmetic (Ortiz-Jiménez et al., 2023).

**The Objective of Model Merging.** Let $D = \{D_t \subset X\}_{t \in [T]}$ be a collection of $T$ datasets, with $\theta_1, \ldots, \theta_T$ representing the parameters of models fine-tuned from $\theta_0$ using the corresponding datasets. The parameters resulting from the merging process are denoted by $\theta_{\text{merge}}$. The objective of model merging is defined as:

$$f(x; \theta_{\text{merge}}) = \begin{cases} f(x; \theta_t) & \text{if } x \in D_t \\ f(x; \theta_0) & \text{if } x \notin \bigcup_{t=1}^{T} D_t \end{cases} \tag{9}$$

Next, we will independently apply the objective to each module that satisfies linearity, using standard weighted operations of task arithmetic, i.e., $\theta_{\text{merge}}^i = \theta_0^i + \sum_{t=1}^{T} \alpha_t^i \tau_t^i$.

**Objectives Function for Each Module**. For a module located at position $i$ (which could be a layer/attention/MLP/head), let $\theta_t^i$ represent the weights of this module from model $t$, and $f(x; \theta_t^i)$ denote the output of this module given input $x$. The merging parameters $\alpha_1^i, \alpha_2^i, \ldots, \alpha_T^i$ for $\theta_1^i, \theta_2^i, \ldots, \theta_T^i$ can be obtained by

$$\alpha_1^i, \alpha_2^i, \ldots, \alpha_T^i = \arg\min \sum_{t=1}^{T} \mathbb{E}_{x \in H_t^i} \left\| f\left(x; \theta_0^i + \sum_{t'=1}^{T} \alpha_{t'}^i \tau_{t'}^i\right) - f(x; \theta_t^i) \right\|^2 \tag{10}$$

where $\tau_t^i = \theta_t^i - \theta_0^i$, $H_t^i$ represent the set of input features of module $i$ corresponding to $D_t$. In this paper, $H_t^i$ is obtained by inputting all $x \in D_t$ into the model $\theta_0$ and collecting the input features corresponding to module $i$, more details please refer to Appendix A.3.3.

If module $i$ satisfies linearity, by applying Property 1 and Property 2, we have

$$\alpha_1^i, \alpha_2^i, \ldots, \alpha_T^i = \arg\min \sum_{t=1}^{T} \mathbb{E}_{x \in H_t^i} \left\| \sum_{t'=1,t'\neq t}^{T} \alpha_{t'}^i \Delta f\left(x;\theta_{t'}^i\right) + (\alpha_t^i - 1)\Delta f\left(x;\theta_t^i\right) \right\|^2 \quad (11)$$

where $\Delta f\left(x;\theta_t^i\right) = f\left(x;\theta_t^i\right) - f(x;\theta_0^i)$

Solving this equation allows us to directly obtain the closed-form solutions for $\alpha_1^i, \alpha_2^i, \ldots, \alpha_T^i$:

$$[\alpha_1^i, \alpha_2^i, ..., \alpha_T^i]^\top = \boldsymbol{A}^{-1}\boldsymbol{b} \quad (12)$$

where $\boldsymbol{A} \subseteq \mathbb{R}^{T \times T}$ and $\boldsymbol{b} \subseteq \mathbb{R}^T$,

$$\boldsymbol{A}_{j,k} = \sum_{t=1}^{T} \boldsymbol{B}_{tjk}^i , \quad \boldsymbol{b}_j = \sum_{t=1}^{T} \boldsymbol{B}_{tjt}^i , \quad (13)$$

and

$$\boldsymbol{B}_{abc}^i = \mathbb{E}_{x \in H_a^i} \sum_{k=1}^{d} \Delta f\left(x;\theta_b^i\right)_k \Delta f\left(x;\theta_c^i\right)_k ,$$
$$\Delta f\left(x;\theta_t^i\right) = f\left(x;\theta_t^i\right) - f(x;\theta_0^i). \quad (14)$$

where $d$ denote the dimension of vector $f\left(x;\theta^i\right)$. The detailed derivation process can be found in the Appendix A.3.1.

We present our merging process at the layer-level decomposition detailed in Algorithm 1. Initially, the input feature sets $H_t^i$ are compiled for all layers by feeding a randomly sampled subset from the task-related datasets into the pre-trained model $\theta_0$. Subsequently, for each layer, we apply Eq.12 to derive the optimal merging weights, which are then utilized to perform linear merging. The adaptation of this algorithm to other granularities is straightforward, as described in Appendix A.3.3.

---

**Algorithm 1** Merging Modules with Linearity at Layer Level

---

**Input:** $T$ Fine-tuned Models with $L$ layers $\theta_1 = \{\theta_1^1, \theta_1^2, ..., \theta_1^L\}, ..., \theta_T = \{\theta_T^1, \theta_T^2, ..., \theta_T^L\}$, Pre-trained Model $\theta_0 = \{\theta_0^1, \theta_0^2, ..., \theta_0^L\}$, Task-Related Datasets $D_1, D_2, ..., D_T$
**Output:** Merged Model $\theta_{merge} = \{\theta_{merge}^1, \theta_{merge}^2, ..., \theta_{merge}^L\}$
**for** $t = 1$ **to** $T$ **do**
    $d_t \leftarrow \{x \in D_t \,|\, x \text{ is sampled randomly}, |d_t| = N\}$    ▷ Sample a small set of task-related data
    $H_t^1 \leftarrow d_t$    ▷ Set the input of first layer
    **for** $i = 1$ **to** $L - 1$ **do**
        $H_t^{i+1} \leftarrow f(x;\theta_0^i), \,\forall\, x \in H_t^i$    ▷ Prepare input feature sets for Eq.12
**for** $i = 1$ **to** $L$ **do**
    $\alpha_1^i, \alpha_2^i, ..., \alpha_T^i \leftarrow$ Eq.12    ▷ Calculate optimal merging weights with Eq.12
    $\theta_{merge}^i \leftarrow \theta_0^i + \sum_{t=1}^{T} \alpha_t^i(\theta_t^i - \theta_0^i)$    ▷ Merge modules from different models linearly

---

# 4 EXPERIMENTS

## 4.1 EXPERIMENTS SETUP

**Models, Datasets and Evaluation Metrics**. We utilize Llama-2-7B and Llama-2-13B as our backbone models. Fine-tuning is conducted across three distinct tasks: mathematics, coding, and translation. For the mathematics task, we employ the GSM8K (Cobbe et al., 2021) training set. For the coding task, we utilize the Code Alpaca (Chaudhary, 2023) dataset. Lastly, for the translation task, we apply the zh↔en dataset from Xu et al. (2024a). During training, we adopt the prompt of the

Table 1: Results of merging fine-tuned models by different methods with Llama-2-13b as the backbone. For each setting, we fully replicate our method three times with different data sample seeds and compute the mean and standard deviation of the three results. The best and second-best results are highlighted in bold and underlined, respectively. For the hyperparameters of other baselines, we conducted a grid search in each merging setting based on the recommendations in the original paper and reported the best results. We only report the average results of different task evaluations here; for more detailed results, please refer to the Appendix A.1.2.

| Methods | Math & Coding | Math & Translate | Coding & Translate | Math & Coding & Translate |
|---|---|---|---|---|
| Fine-tuned Model | 34.02 | 65.16 | 51.26 | 50.14 |
| Weight Avg | 33.57 | 64.57 | 52.28 | 46.95 |
| DARE | 32.97 | 62.47 | **54.33** | 47.51 |
| Task Arithmetic | 34.60 | 64.72 | 53.22 | 48.24 |
| Ours Layer Level | $34.47_{\pm 0.17}$ | $66.08_{\pm 0.14}$ | $52.42_{\pm 0.15}$ | $\underline{50.80}_{\pm 0.17}$ |
| Ours Attn/MLP Level | $\mathbf{35.11}_{\pm 0.29}$ | $\mathbf{66.11}_{\pm 0.11}$ | $52.51_{\pm 0.26}$ | $\mathbf{51.05}_{\pm 0.16}$ |

FastChat (Zheng et al., 2023) template and fine-tune the models for 3 epochs with a batch size of 128 and a learning rate of $2 \times 10^{-5}$.

For evaluation, we test mathematical capability using the GSM8K (Cobbe et al., 2021) test set, coding capability with the HumanEval (Chen et al., 2021), and translation capability with the tools and datasets from (Xu et al., 2024a).

**Algorithm Implementation Details**. In practice, we observed varying levels of bias in the feature magnitudes across different datasets. To mitigate potential adverse impact, we implemented a mechanism to balance the influence of each sample, thereby resulting in a minor modification of the original formula. For further details, please refer to the Appendix A.3.2. Due to the constraints of our compact computational and storage budget, we use a limited dataset of only 30 samples per task for calculating the merging weights. Surprisingly, this minimal data size demonstrates a commendable performance, underscoring the efficiency of our method in terms of data requirements and computational costs.

## 4.2 MAIN RESULTS

We compare our proposed method with several baseline approaches: **Weight Average** (Wortsman et al., 2022a;b): Weight Average is the simplest and most direct model merging method, which involves taking the average of the parameters of fine-tuned models. **Task Arithmetic**(Ilharco et al., 2023): Task arithmetic entails a straightforward weighted combination of the *task vectors* and weight of the pre-trained model. **DARE**(Yu et al., 2024): DARE builds upon task arithmetic by employing random dropout of parameters within the *task vectors* to reduce conflicts between different *task vectors* during the merging process.

We present the results of merging fine-tuned models employing various methodologies, utilizing Llama-2-13b and Llama-2-7b (Touvron et al., 2023) as the fine-tuning backbones. The findings are detailed in Table 1 and Table 2, respectively. Each value indicates the average evaluation results across multiple related tasks. It is evident that our approach surpasses the baseline in most merging settings, particularly in the case of merging three models, where a performance improvement of 3% is observed, albeit slightly lower results were achieved in the merging of code and translation models. This validates the effectiveness of our method. Furthermore, the similar outcomes on both Llama-2-7b and Llama-2-13b demonstrate that our method has consistent performance.

## 4.3 ABLATION AND DISCUSSION

Since the optimal merging weights are derived through a closed-form solution, our proposed merging method lacks adjustable hyperparameters. Consequently, our discussion centers on the influence

Table 2: Results of merging fine-tuned models by different methods with Llama-2-7b as the backbone. The details of this table are the same as Table 1. For more results, please refer to the Appendix A.1.2.

| Methods | Math & Coding | Math & Translate | Coding & Translate | Math & Coding & Translate |
|---|---|---|---|---|
| Fine-tuned Model | 32.35 | 62.62 | 50.99 | 48.65 |
| Weight Avg | 25.14 | 53.70 | 49.25 | 38.30 |
| DARE | 19.11 | 50.77 | 50.15 | 39.00 |
| Task Arithmetic | **27.45** | 54.65 | **51.11** | 39.59 |
| Ours Layer Level | 27.18$_{\pm 0.04}$ | 49.79$_{\pm 0.43}$ | 50.64$_{\pm 0.64}$ | 42.68$_{\pm 0.26}$ |
| Ours Attn/MLP Level | 27.33$_{\pm 0.10}$ | **55.86**$_{\pm 0.48}$ | 49.80$_{\pm 0.23}$ | **42.88**$_{\pm 0.27}$ |

Table 3: Ablation results of merging fine-tuned models by different levels in our method. For each setting, we fully replicate our method three times with different seeds and compute the mean and standard deviation of the three results.

| Level | Math & Coding | Math & Translate | Coding & Translate | Math & Coding & Translate |
|---|---|---|---|---|
| | Llama-2-7B | | | |
| Ours Model Level | 28.34$_{\pm 0.22}$ | 13.38$_{\pm 0.57}$ | 46.51$_{\pm 0.15}$ | 35.95$_{\pm 0.51}$ |
| Ours Layer Level | 27.18$_{\pm 0.04}$ | 49.79$_{\pm 0.43}$ | 50.64$_{\pm 0.64}$ | 42.68$_{\pm 0.26}$ |
| Ours Attn/MLP Level | 27.33$_{\pm 0.10}$ | 55.86$_{\pm 0.48}$ | 49.80$_{\pm 0.23}$ | 42.88$_{\pm 0.27}$ |
| Ours Head/MLP Level | 13.08$_{\pm 9.33}$ | 14.17$_{\pm 1.14}$ | 41.14$_{\pm 12.39}$ | 8.16$_{\pm 1.30}$ |
| | Llama-2-13B | | | |
| Ours Model Level | 31.27$_{\pm 0.43}$ | 18.90$_{\pm 2.45}$ | 15.75$_{\pm 7.60}$ | 50.68$_{\pm 0.10}$ |
| Ours Layer Level | 34.47$_{\pm 0.17}$ | 66.08$_{\pm 0.14}$ | 52.42$_{\pm 0.15}$ | 50.80$_{\pm 0.17}$ |
| Ours Attn/MLP Level | 35.11$_{\pm 0.29}$ | 66.11$_{\pm 0.11}$ | 52.51$_{\pm 0.26}$ | 51.05$_{\pm 0.16}$ |
| Ours Head/MLP Level | 12.91$_{\pm 15.60}$ | 30.17$_{\pm 19.36}$ | 13.71$_{\pm 3.50}$ | 22.59$_{\pm 19.65}$ |

of decomposition granularity on our approach. In addition to the Layer-Level and Attention/MLP-Level configurations previously illustrated, we also apply our method at the Model-Level, which serves as a baseline setting without any decomposition, as well as at the Head/MLP-Level, where attentions are further vertically divided into individual heads. [8]

**Results at the Model Level.** We also attempted to directly apply our method for model merging at the model level. The results in Table 3 indicate that, in most cases, it failed to yield reasonable outcomes. A visualization of the parameters calculated at the model level can be found in Table 13. The failure at the model level can be attributed to the lack of sufficient linear correlation between the weight differences caused by fine-tuning and the differences in output features resulting from fine-tuning. The absence of linearity prevents us from leveraging feature-level computations to reasonably merge parameters, ultimately leading to unpredictable results. This implies that our method is only effective under linear conditions.

**Results at the Head/MLP Level.** Despite exhibiting good linearity at the Head level, the merging with our method at this level reveals unstable outcomes, shown in Table 3. We also provide a visual example of the parameters during actual Head level merging in Figure 5.

---

[8]For the Head/MLP-level, due to the architecture of the transformer, the output features corresponding to a single head cannot be directly obtained. Therefore, we employed certain techniques to achieve this in practice. Please refer to Appendix A.3.5 for additional results and analysis.

We offer a conjectural explanation for the occurrence of this situation here. Many studies (Wang et al., 2023; Chen et al., 2024; Zhang et al., 2024) indicate that each head in the model serves different functions during inference. After fine-tuning various tasks, certain heads may take on distinct roles, while our objective function may ask the head to simultaneously average close to multiple heads with different functionalities. This requirement can be quite challenging for an individual head, ultimately leading to a collapse of functionality of the merged head.

However, this may not indicate that the linearity of the head is unhelpful for the merging process. Considering the specific functionality of the head (or the particular task associated with that head) during the merging to redefine a more reasonable merging objective could be a valuable direction for future research.

## 5 RELATED WORK

**Weight Interpolation and Task Arithmetic.** In recent years, the increasing scale of large language models has significantly heightened the demand for data and training costs associated with fine-tuning multi-task models. To merge the capabilities of various existing single-task models into a unified framework, several model merging techniques (Daheim et al., 2024; Jin et al., 2023; Wan et al., 2024; Yang et al., 2024b) have been developed. The simplest and most intuitive approach is weight interpolation(Frankle et al., 2020; Izmailov et al., 2018; Ramé et al., 2023; 2022), which has been applied to enhance model generalization(Wortsman et al., 2022b), improve specific single-task performance(Wortsman et al., 2022a), and boost multi-task effectiveness(Ilharco et al., 2022; Li et al., 2022). Subsequently, Task Arithmetic(Ilharco et al., 2023) was introduced, enabling the merging of multiple models through a weighted operation on the parameter difference $\tau$. Many subsequent methods(Yang et al., 2024c; Yu et al., 2024; Yadav et al., 2023; Zhang et al., 2023) have been proposed based on the principles of Task Arithmetic.

**Linearity and Task Arithmetic.** The research conducted by Ortiz-Jiménez et al. (2023) begins with the relevant theories of the Neural Tangent Kernel (NTK)(Jacot et al., 2018; Chizat et al., 2019) to explore the connection between linearity and task arithmetic. Although experimental validations indicate that fine-tuned models do not comply with NTK's conclusion regarding the approximate one-order Taylor expansion of models, Ortiz-Jiménez et al. (2023) observes that linearity still can aid task arithmetic. To enhance model linearity, Ortiz-Jiménez et al. (2023) proposes a constraint on the parameter update space during fine-tuning, thereby fine-tuning a model that achieves linearity, ultimately improving the efficiency of model merging. To expedite this training process, various studies have attempted to linearize only a subset of parameters during training. For instance, Jin et al. (2024) suggest linearly fine-tuning solely the linear layers in the attention modules. Tang et al. (2024) combine linearly fine-tuning with Parameter-Efficient Fine-Tuning (PEFT) techniques by linearizing only the Adapter modules, thereby reducing computational costs. Liu et al. (2024) derives a closed-form linearized solution for efficiently fine-tuning Transformer networks.

In conclusion, modifying fine-tuning methods to achieve a more linear model can reduce interference during model merging. However, these methods still require training data and retraining, which is often challenging in practical scenarios.

## 6 CONCLUSION

In this study, we performed a statistical analysis to investigate the linearity of model-level and decomposed submodule-level components. Our findings indicate that the missing linearity in full models can be found in the submodules. This observation motivated us to develop an innovative, training-free approach to enhance task arithmetic performance. Specifically, we first decompose the model into its constituent submodules and then compute the optimal merging weights for each module based on a closed-form solution derived from linear properties. Subsequently, we perform a linear merging of all submodules using the corresponding merging weights. Our experimental results demonstrate that this approach significantly outperforms various baseline techniques, including standard task arithmetic methods, across different model scales and diverse task scenarios. These findings underscore the advantages of leveraging the linearity of submodules and present a novel perspective for investigating effective and practical solutions for multi-task model merging.

ACKNOWLEDGEMENTS

This work was supported in part by NSFC No. 62222117.

ETHICS STATEMENT

We proposed a novel approach to enhance model merging performance in large language models (LLMs). Our methodology utilizes publicly available language datasets and leverages pre-trained language models for experimental validation. We do not believe that our code or method is inherently subject to concerns such as discrimination, bias, fairness, inappropriate potential applications, impact, privacy and security issues, legal compliance, or research integrity concerns. However, language datasets and models may possess intrinsic biases that could be inherited by models merged using our approach.

REPRODUCIBILITY STATEMENT

To ensure the reproducibility of our approach, we provide key information from the main text and Appendix as follows.

**Algorithm**. We provide our algorithm in Algorithm 1.

**Experimental Details**. We provide our experimental details in Section 4. Moreover, we provide detailed experiment settings in Appendices A.2 A.3.1, A.3.2, A.3.3 and A.3.5.

Moreover, we are committed to providing the source code of our approach, if accepted.

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

# A APPENDIX

## A.1 MORE RESULTS

### A.1.1 MORE RESULTS IN ANALYZING THE LINEARITY OF SINGLE FINE-TUNED MODELS

To ascertain whether the previously mentioned observations regarding linearity remain applicable within the context of model merging, we followed the methodology outlined in Zhou et al. (2024). We analyzed two types of linearity metrics introduced in their work, cosine similarity and projection distance in feature space.

Specifically, we analyze the cosine similarity between two sets of feature deltas: the differences between the output features of the merged module (derived from multiple fine-tuned models) and those of the pre-trained module, denoted as $\Delta f(x; \theta_0^i + \sum_{t=1}^{T} \alpha_t^i \tau_t^i) = f(x; \theta_0^i + \sum_{t=1}^{T} \alpha_t^i \tau_t^i) - f(x; \theta_0^i)$, as well as the corresponding weighted combination of the differences in output features from the fine-tuned models, represented as $\sum_{t=1}^{T} \alpha_t^i \Delta f(x; \theta_0^i + \tau_t^i) = \sum_{t=1}^{T} \alpha_t (f(x; \theta_0^i + \tau_t^i) - f(x; \theta_0^i))$. This similarity is denoted by the following equation:

$$\text{cosine}_{merge}(x; \alpha_{1...T}^i) = \cos\left(\Delta f(x; \theta_0^i + \sum_{t=1}^{T} \alpha_t^i \tau_t^i), \sum_{t=1}^{T} \alpha_t^i \Delta f(x; \theta_0^i + \tau_t^i)\right) \quad (15)$$

and

$$\text{cosine}_{merge}^{\mathbb{E}}(\alpha_{1...T}^i) = \mathbb{E}_{x \in H^i} \text{cosine}_{merge}(x; \alpha_{1...T}^i). \quad (16)$$

Following Zhou et al. (2024), we compare these cosine similarities against a natural baseline: the average cosine similarity of the feature deltas induced by fine-tuning across all possible pairs of the tuned models, denoted as:

$$\text{cosine}_{base}^{\mathbb{E}} = \mathbb{E}_{x \in H^i} \frac{1}{T(T-1)} \sum_{1 \le t < t' \le T} \cos(\Delta f(x; \theta_0^i + \tau_t^i), \Delta f(x; \theta_0^i + \tau_{t'}^i)) \quad (17)$$

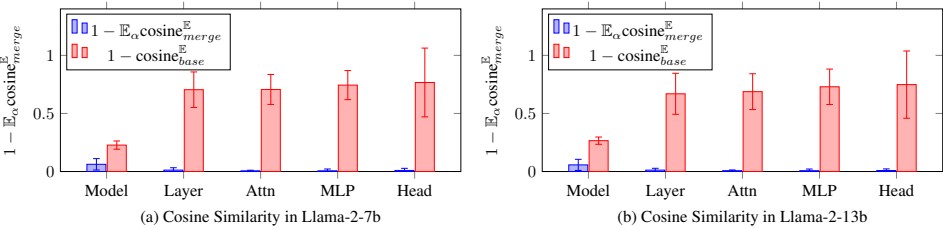

(a) Cosine Similarity in Llama-2-7b      (b) Cosine Similarity in Llama-2-13b

Figure 4: Comparison of Cosine Similarity in full models and submodules within the context of model merging. A lower value indicates better linearity. For each backbone, we computed the results across all possible combinations of three fine-tuned models, alongside the corresponding merging weights $\alpha_1$ and $\alpha_2$ (and $\alpha_3$). In the case of merging two models, there are three combinations of fine-tuned models, yielding 25 possible configurations for merging weights $\alpha_t \in [0.2, 0.4, 0.6, 0.8, 1]$ where $t \in [1, 2]$. For merging three models, there's only one combination of fine-tuned models, resulting in 27 configurations for merging weights $\alpha_t \in [0.3, 0.5, 0.7]$ with $t \in [1, 2, 3]$. Error bars represent standard deviations.

**Results of Cosine Similarity.** As illustrated in Figure 4 (a) and (b), we observe cosine similarity results consistent with Zhou et al. (2024), revealing a significant disparity between $1 - \mathbb{E}_\alpha \text{cosine}_{merge}^{\mathbb{E}}$ and the baseline $1 - \text{cosine}_{base}^{\mathbb{E}}$. Furthermore, a greater gap and reduced values are noted at the module level compared to the model level, suggesting a more favorable condition for linear merging.

### A.1.2 MORE RESULTS IN MERGING MODULES WITH LINEARITY

In this section, we report more detailed results of Tables 1 and 2 in Tables 4 and 5, as well as the outcomes of our approach on additional models and datasets. The results of model merging on the

Qwen-2-0.5B(Yang et al., 2024a) model, along with datasets Gsm8k(Cobbe et al., 2021), Cosmos QA (Huang et al., 2019), Ropes (Lin et al., 2019), and Winogrande (Sakaguchi et al., 2020), are presented in Table 6. We also report result on Qwen-2-7B(Qwen Team, 2024) model. During training, we adopt the prompt template from FastChat (Zheng et al., 2023), fine-tuning the models for 3 epochs with a batch size of 128 and a learning rate of $2 \times 10^{-5}$.

Table 4: Detailed experimental results on LLama-2-13B.

| Method Llama-2-13B | Merging Two Models | | | | | |
|---|---|---|---|---|---|---|
| | Math | Coding | Avg | Math | Translate | Avg |
| Fine-tuned Model | 47.91 | 20.12 | 34.02 | 47.91 | 82.40 | 65.16 |
| Weight Avg | 42.76 | 24.39 | 33.57 | 45.11 | **84.04** | 64.57 |
| DARE | 41.55 | **24.39** | 32.97 | 40.94 | 83.99 | 62.47 |
| Task Arithmetic | 45.41 | 23.78 | **34.60** | 45.87 | 83.58 | 64.72 |
| Ours Model Level | **48.12**$_{\pm0.29}$ | 14.43$_{\pm0.57}$ | 31.27$_{\pm0.43}$ | 0.20$_{\pm0.09}$ | 37.60$_{\pm4.91}$ | 18.90$_{\pm2.45}$ |
| Ours Layer Level | 46.37$_{\pm0.34}$ | 22.56$_{\pm0.00}$ | 34.47$_{\pm0.17}$ | 48.42$_{\pm0.25}$ | 83.75$_{\pm0.04}$ | 66.08$_{\pm0.14}$ |
| Ours Attn/MLP Level | 46.85$_{\pm0.00}$ | 23.37$_{\pm0.57}$ | 35.11$_{\pm0.29}$ | **48.55**$_{\pm0.20}$ | 83.68$_{\pm0.03}$ | **66.11**$_{\pm0.11}$ |
| Ours Head/MLP Level | 15.87$_{\pm20.86}$ | 9.96$_{\pm10.45}$ | 12.91$_{\pm15.60}$ | 16.27$_{\pm17.88}$ | 44.07$_{\pm20.93}$ | 30.17$_{\pm19.36}$ |

| Method Llama-2-13B | Merging Two Models | | | Merging Three Models | | | |
|---|---|---|---|---|---|---|---|
| | Coding | Translate | Avg | Math | Coding | Translate | Avg |
| Fine-tuned Model | 20.12 | 82.40 | 51.26 | 47.91 | 20.12 | 82.40 | 50.14 |
| Weight Avg | 20.12 | 84.43 | 52.28 | 33.13 | 23.78 | **83.95** | 46.95 |
| DARE | **24.39** | 84.27 | **54.33** | 34.87 | 23.78 | 83.88 | 47.51 |
| Task Arithmetic | 21.95 | **84.48** | 53.22 | 38.36 | 22.56 | 83.80 | 48.24 |
| Ours Model Level | 0.00$_{\pm0.00}$ | 31.50$_{\pm15.19}$ | 15.75$_{\pm7.60}$ | **47.08**$_{\pm0.68}$ | 23.58$_{\pm0.29}$ | 81.38$_{\pm0.43}$ | 50.68$_{\pm0.10}$ |
| Ours Layer Level | 20.53$_{\pm0.29}$ | 84.31$_{\pm0.02}$ | 52.42$_{\pm0.15}$ | 45.21$_{\pm0.26}$ | 23.37$_{\pm0.29}$ | 83.81$_{\pm0.04}$ | 50.80$_{\pm0.17}$ |
| Ours Attn/MLP Level | 20.73$_{\pm0.50}$ | 84.29$_{\pm0.04}$ | 52.51$_{\pm0.26}$ | 45.59$_{\pm0.09}$ | **23.78**$_{\pm0.50}$ | 83.78$_{\pm0.01}$ | **51.05**$_{\pm0.16}$ |
| Ours Head/MLP Level | 0.00$_{\pm0.00}$ | 27.43$_{\pm7.00}$ | 13.71$_{\pm3.50}$ | 14.99$_{\pm20.24}$ | 8.13$_{\pm11.07}$ | 44.66$_{\pm27.65}$ | 22.59$_{\pm19.65}$ |

Table 5: Detailed experimental results on LLama-2-7B.

| Method Llama-2-7B | Merging Two Models | | | | | |
|---|---|---|---|---|---|---|
| | Math | Coding | Avg | Math | Translate | Avg |
| Fine-tuned Model | 43.97 | 20.73 | 32.35 | 43.97 | 81.26 | 62.62 |
| Weight Avg | 31.99 | 18.29 | 25.14 | 25.02 | **82.38** | 53.70 |
| DARE | 21.15 | 17.07 | 19.11 | 19.41 | 82.14 | 50.77 |
| Task Arithmetic | 37.83 | 17.07 | 2745 | 27.07 | 82.24 | 54.65 |
| Ours Model Level | **43.47**$_{\pm0.50}$ | 13.21$_{\pm0.76}$ | **28.34**$_{\pm0.22}$ | 2.27$_{\pm0.34}$ | 24.48$_{\pm0.80}$ | 13.38$_{\pm0.57}$ |
| Ours Layer Level | 35.25$_{\pm0.21}$ | **19.11**$_{\pm0.29}$ | 27.18$_{\pm0.04}$ | 17.77$_{\pm0.84}$ | 81.82$_{\pm0.03}$ | 49.79$_{\pm0.43}$ |
| Ours Attn/MLP Level | 35.96$_{\pm0.09}$ | 18.70$_{\pm0.29}$ | 27.33$_{\pm0.10}$ | **30.00**$_{\pm0.98}$ | 81.73$_{\pm0.03}$ | **55.86**$_{\pm0.48}$ |
| Ours Head/MLP Level | 17.41$_{\pm13.21}$ | 8.74$_{\pm6.57}$ | 13.08$_{\pm9.33}$ | 0.58$_{\pm0.72}$ | 27.77$_{\pm1.57}$ | 14.17$_{\pm1.14}$ |

| Method Llama-2-7B | Merging Two Models | | | Merging Three Models | | | |
|---|---|---|---|---|---|---|---|
| | Coding | Translate | Avg | Math | Coding | Translate | Avg |
| Fine-tuned Model | 20.73 | 81.26 | 50.99 | 43.97 | 20.73 | 81.26 | 48.65 |
| Weight Avg | 15.85 | 82.64 | 49.25 | 17.66 | 15.24 | **81.99** | 38.30 |
| DARE | 17.68 | 82.62 | 50.15 | 19.64 | 15.24 | 82.13 | 39.00 |
| Task Arithmetic | **19.51** | **82.70** | **51.11** | 22.29 | 14.63 | 81.86 | 39.59 |
| Ours Model Level | 11.79$_{\pm0.29}$ | 81.23$_{\pm0.18}$ | 46.51$_{\pm0.15}$ | 41.57$_{\pm0.62}$ | 14.43$_{\pm0.57}$ | 51.84$_{\pm1.54}$ | 35.95$_{\pm0.51}$ |
| Ours Layer Level | 18.90$_{\pm1.32}$ | 82.38$_{\pm0.03}$ | 50.64$_{\pm0.64}$ | 30.50$_{\pm0.54}$ | **15.85**$_{\pm0.50}$ | 81.68$_{\pm0.05}$ | 42.68$_{\pm0.26}$ |
| Ours Attn/MLP Level | 17.28$_{\pm0.57}$ | 82.33$_{\pm0.12}$ | 49.80$_{\pm0.23}$ | **32.02**$_{\pm0.31}$ | 14.84$_{\pm0.57}$ | 81.79$_{\pm0.03}$ | **42.88**$_{\pm0.27}$ |
| Ours Head/MLP Level | 16.06$_{\pm2.74}$ | 66.22$_{\pm22.41}$ | 41.14$_{\pm12.39}$ | 0.00$_{\pm0.00}$ | 0.00$_{\pm0.00}$ | 24.48$_{\pm3.90}$ | 8.16$_{\pm1.30}$ |

### A.1.3 ABLATION STUDY ON NUMBER OF MODEL TO BE MERGED

To demonstrate the advantages of our method in scenarios where more models are merged simultaneously, we increase the number of task-specific models merged simultaneously gradually and calculate the average performance of the merged models.

Table 6: Detailed experimental results on Qwen-2-0.5B.

| Method Qwen-2-0.5B | Merging Three Models | | | | | | | |
|---|---|---|---|---|---|---|---|---|
| | gsm8k | cosmos_qa | ropes | Avg | gsm8k | cosmos_qa | winogrande | Avg |
| Fine-tuned Model | 31.61 | 65.01 | 45.65 | 47.42 | 31.61 | 65.01 | 56.59 | 51.07 |
| Weight Avg | 25.09 | 51.91 | 37.50 | 38.17 | 17.74 | 49.31 | 51.82 | 39.62 |
| DARE | 34.57 | 40.09 | 41.61 | 38.76 | 31.24 | 45.14 | 51.34 | 42.57 |
| Task Arithmetic | 26.54 | 50.83 | 37.56 | 38.31 | 22.52 | 48.63 | 51.82 | 40.99 |
| Ours Model Level | $31.35_{\pm 0.11}$ | $22.71_{\pm 2.15}$ | $\mathbf{46.95}_{\pm 0.09}$ | $33.67_{\pm 0.65}$ | $14.56_{\pm 0.00}$ | $47.95_{\pm 0.00}$ | $\mathbf{52.96}_{\pm 0.00}$ | $38.49_{\pm 0.00}$ |
| Ours Layer Level | $30.52_{\pm 0.11}$ | $51.76_{\pm 0.05}$ | $38.07_{\pm 0.03}$ | $\mathbf{40.12}_{\pm 0.01}$ | $\mathbf{27.71}_{\pm 0.04}$ | $51.00_{\pm 0.03}$ | $52.31_{\pm 0.00}$ | $43.67_{\pm 0.02}$ |
| Ours Attn/MLP Level | $\mathbf{30.55}_{\pm 0.08}$ | $\mathbf{52.07}_{\pm 0.08}$ | $37.62_{\pm 0.06}$ | $40.08_{\pm 0.03}$ | $27.48_{\pm 0.27}$ | $\mathbf{51.13}_{\pm 0.07}$ | $52.55_{\pm 0.00}$ | $\mathbf{43.72}_{\pm 0.11}$ |
| Ours Head/MLP Level | $29.19_{\pm 0.83}$ | $50.07_{\pm 1.98}$ | $37.53_{\pm 0.33}$ | $38.93_{\pm 1.05}$ | $15.05_{\pm 12.09}$ | $39.77_{\pm 12.24}$ | $50.57_{\pm 0.61}$ | $35.13_{\pm 8.31}$ |

| Method Qwen-2-0.5B | Merging Three Models | | | | | | | |
|---|---|---|---|---|---|---|---|---|
| | gsm8k | winogrande | ropes | Avg | winogrande | cosmos_qa | ropes | Avg |
| Fine-tuned Model | 31.61 | 56.59 | 45.65 | 44.62 | 56.59 | 65.01 | 45.65 | 55.75 |
| Weight Avg | 10.84 | 50.93 | $\mathbf{49.09}$ | 36.96 | 51.42 | 44.36 | 45.23 | 47.00 |
| DARE | $\mathbf{33.97}$ | 52.39 | 48.85 | $\mathbf{45.07}$ | 51.82 | 42.91 | 46.20 | 46.97 |
| Task Arithmetic | 15.54 | 52.06 | 48.67 | 38.76 | 51.90 | 44.84 | $\mathbf{45.35}$ | 47.36 |
| Ours Model Level | $7.77_{\pm 0.11}$ | $\mathbf{53.68}_{\pm 0.00}$ | $31.34_{\pm 0.06}$ | $30.93_{\pm 0.02}$ | $\mathbf{52.39}_{\pm 0.00}$ | $48.00_{\pm 0.05}$ | $30.95_{\pm 0.45}$ | $43.78_{\pm 0.13}$ |
| Ours Layer Level | $24.30_{\pm 0.19}$ | $51.46_{\pm 0.04}$ | $48.79_{\pm 0.06}$ | $\mathbf{41.52}_{\pm 0.03}$ | $51.78_{\pm 0.04}$ | $50.10_{\pm 0.02}$ | $43.21_{\pm 0.27}$ | $\mathbf{48.36}_{\pm 0.08}$ |
| Ours Attn/MLP Level | $24.18_{\pm 0.45}$ | $51.94_{\pm 0.04}$ | $48.25_{\pm 0.00}$ | $41.46_{\pm 0.17}$ | $51.62_{\pm 0.04}$ | $\mathbf{50.64}_{\pm 0.08}$ | $42.48_{\pm 0.03}$ | $48.25_{\pm 0.05}$ |
| Ours Head/MLP Level | $24.00_{\pm 0.27}$ | $51.58_{\pm 0.08}$ | $47.98_{\pm 0.21}$ | $41.18_{\pm 0.04}$ | $50.93_{\pm 0.16}$ | $50.22_{\pm 0.20}$ | $43.21_{\pm 0.03}$ | $48.12_{\pm 0.02}$ |

Table 7: Experimental results on LLaMA-2-7B with More Dataset: Winogrande, Cosmos qa and Ropes

| Method Llama-2-7b | Merging Two Models | | | | | | | | |
|---|---|---|---|---|---|---|---|---|---|
| | Winogrande | Cosmos qa | Avg | Winogrande | Ropes | Avg | Cosmos qa | Ropes | Avg |
| Fine-tuned Model | 82.26 | 84.72 | 83.49 | 82.26 | 75.42 | 78.84 | 84.72 | 75.42 | 80.07 |
| Task Arithmetic | 72.22 | 71.85 | 72.03 | 72.14 | 54.28 | 63.21 | 73.48 | 58.75 | 66.11 |
| Ours Attn/MLP Level | 68.42 | 84.18 | 76.3 | 69.47 | 69.98 | 69.72 | 73.07 | 68.59 | 70.83 |

| Method Llama-2-7b | **Merging Three Models** | | | |
|---|---|---|---|---|
| | Winogrande | Cosmos qa | Ropes | Avg |
| Fine-tuned Model | 82.26 | 84.72 | 75.42 | 80.8 |
| Task Arithmetic | 71.25 | 64.13 | 55.85 | 63.7433 |
| Ours Attn/MLP Level | 71.49 | 68.70 | 56.70 | 65.63 |

In the table 10, we can observe that as the number of models increases, our method gradually surpasses Task Arithmetic and the performance gap becomes larger, which demonstrates the efficiency of our method particularly when more models are merged simultaneously.

### A.1.4 ABLATION ON THE NUMBER OF DATA

From the table 11, we can see that our method shows consistent performance across different data quantities, sometimes achieving good results with as few as 3 data per task. As a result, if one wishes to use our method in a situation of absolutely no data, we believe that simply synthesizing a few data for the corresponding task through manual methods or with the help of LLMs is feasible.

### A.2 MORE DETAILS ABOUT NON-LINEARITY SCORE

Overall, we achieve the practical implementation of the Non-linearity Score through discretization and using the Euclidean distance as distance metric $\mathcal{D}$ in the feature space.

Given a fine-tuned model $\theta$, we interpolate between this model and its corresponding pre-trained model $\theta_0$ using $k \in \{0, 1, 2, \ldots, N-1, N\}$. This results in $N+1$ models $\theta_k = \frac{k}{N}\theta + \frac{1-k}{N}\theta_0 = \theta_0 + \frac{k}{N}(\theta - \theta_0)$. If $\theta$ satisfies linearity in Definition 1 under a distance metric $\mathcal{D}$ in the feature space, then for any $i, j \in \{0, 1, 2, \ldots, N-1, N\}$, the following approximation holds:

$$\mathcal{D}(f(x; \theta_i), f(x; \theta_j)) \approx \frac{(i-j)}{N}\mathcal{D}(f(x; \theta), f(x; \theta_0)) \tag{18}$$

Table 8: Experimental results on Qwen-2.5-7B

| Method | | | | Merging Two Models | | | | | | Merging Three Models | | | |
|---|---|---|---|---|---|---|---|---|---|---|---|---|---|
| Qwen-2.5-7B | Math | Coding | Avg | Math | Translate | Avg | Coding | Translate | Avg | Math | Coding | Translate | Avg |
| Fine-tuned Model | 71.49 | 60.36 | 65.92 | 71.49 | 85.83 | 78.66 | 60.36 | 85.83 | 73.09 | 71.49 | 60.36 | 85.83 | 72.56 |
| Task Arithmetic | 75.13 | 67.68 | 71.40 | 78.69 | 86.43 | 82.56 | 64.02 | 86.30 | 75.16 | 78.92 | 65.24 | 86.56 | 76.90 |
| Ours Layer Level | 78.01 | 64.02 | 71.01 | 77.93 | 86.55 | 82.24 | 64.63 | 86.59 | 75.61 | 81.04 | 64.02 | 86.65 | 77.23 |
| Ours Attn/MLP Level | 77.86 | 64.63 | 71.24 | 78.01 | 86.58 | 82.29 | 64.02 | 86.62 | 75.32 | 80.66 | 65.24 | 86.64 | 77.51 |

Table 9: Experimental results with more baseline

| Method | | | | Merging Two Models | | | | | | Merging Three Models | | | |
|---|---|---|---|---|---|---|---|---|---|---|---|---|---|
| Llama-2-7B | Math | Coding | Avg | Math | Translate | Avg | Coding | Translate | Avg | Math | Coding | Translate | Avg |
| Fine-tuned Model | 43.97 | 20.73 | 32.35 | 43.97 | 81.26 | 62.62 | 20.73 | 81.26 | 50.99 | 43.97 | 20.73 | 81.26 | 48.65 |
| Weight Avg | 31.99 | 18.29 | 25.14 | 25.02 | 82.38 | 53.7 | 15.85 | 82.64 | 49.25 | 17.66 | 15.24 | 81.99 | 38.3 |
| DARE | 21.15 | 17.07 | 19.11 | 19.41 | 82.14 | 50.77 | 17.68 | 82.62 | 50.15 | 19.64 | 15.24 | 82.13 | 39 |
| Task Arithmetic | 37.83 | 17.07 | 27.45 | 27.07 | 82.24 | 54.65 | 19.51 | 82.7 | 51.11 | 22.29 | 14.63 | 81.86 | 39.59 |
| Breadcrumbs | 18.11 | 18.90 | 18.51 | 16.22 | 82.24 | 49.23 | 19.51 | 82.26 | 50.89 | 7.96 | 18.90 | 82.78 | 36.55 |
| TIES | 34.49 | 18.90 | 26.69 | 32.97 | 82.11 | **57.54** | 19.51 | 82.75 | **51.13** | 29.03 | 15.85 | 81.49 | 42.12 |
| Consensus TA | 27.21 | 17.68 | 22.44 | 25.39 | 82.47 | 53.93 | 18.29 | 82.51 | 50.40 | 26.38 | 17.68 | 82.36 | 42.14 |
| Ours Layer Level | 35.25 | 19.11 | 27.18 | 17.77 | 81.82 | 49.79 | 18.90 | 82.38 | 50.64 | 30.5 | 15.85 | 81.68 | 42.68 |
| Ours Attn/MLP Level | 35.96 | 18.70 | **27.33** | 30.00 | 81.73 | 55.86 | 17.28 | 82.33 | 49.8 | 32.02 | 14.84 | 81.79 | **42.88** |

Since $f(x; \theta)$ is a vector, we utilize the Euclidean distance as distance metric $\mathcal{D}$, to measure the distance between features. To assess the discrepancy between the actual situation regarding $\theta$ and the Eq.18, the Non-linearity Score is defined as follows:

$$\text{Non-linearity Score} = \sum_{i \in \{0,..,N\}} \sum_{j \in \{0,..,N\}} \left( \frac{\mathcal{D}(f(x;\theta_i), f(x;\theta_j))}{\mathcal{D}(f(x;\theta), f(x;\theta_0))} - \frac{|i-j|}{N} \right)^2 \quad (19)$$

We set $N = 10$ in this paper. Some visualization of $\frac{\mathcal{D}(f(x;\theta_i), f(x;\theta_j))}{\mathcal{D}(f(x;\theta), f(x;\theta_0))}$ is shown in Figure 6.

We can also extend the Non-linearity Score for analyzing the linearity of merging multiple fine-tuned models, maintaining the same functionality as the Projection Distance metric discussed in Section 2.3. Due to the limitations of our compact computational and storage budget, we employ Projection Distance in Section 2.3 for its higher efficiency.

## A.3 MORE DETAILS ABOUT OUR PROPOSED METHOD

### A.3.1 THE DERIVATION OF THE CLOSED-FORM SOLUTION

**Objectives Function**. For a module located at position $i$ (which could be a layer/attention/MLP), let $\theta_t^i$ represent the weights of this module from model $t$, and $f(x; \theta_t^i)$ denote the output of this module given input $x$. The merging parameters $\alpha_1^i, \alpha_2^i, \ldots, \alpha_T^i$ for $\theta_1^i, \theta_2^i, \ldots, \theta_T^i$ can be obtained by

$$\alpha_1^i, \alpha_2^i, \ldots, \alpha_T^i = \arg\min \sum_{t=1}^{T} \mathbb{E}_{x \in H_t^i} \left\| f\left( x; \theta_0^i + \sum_{t'=1}^{T} \alpha_{t'}^i \tau_{t'}^i \right) - f(x; \theta_t^i) \right\|^2 \quad (20)$$

where $\tau_t^i = \theta_t^i - \theta_0^i$, $H_t^i$ represent the set of input features of module $i$ corresponding to $D_t$. In this paper, $H_t^i$ is obtained by inputting all $x \in D_t$ into the model $\theta_0$ and collecting the input features corresponding to module $i$.

If module $i$ satisfies linearity, by applying Property 2, we have

Table 10: Ablation study on Number of Model to be Merged

| Avg. Acc | Coding & Translate | Coding & Translate & Cosmos_qa | Coding & Translate & Cosmos_qa & Ropes | Coding & Translate & Cosmos_qa & Ropes & Winogrande |
|---|---|---|---|---|
| Task Arithmetic | 0.5111 | 0.5643 | 0.5257 | 0.4666 |
| Ours | 0.4980 | 0.5581 | 0.5382 | 0.5489 |
| Gap $\Delta$ | -2.6% | -1.1% | +2.3% | +14.9% |

Table 11: Ablation on the Number of Data

| | Data Num per Task | 1 | 3 | 10 | 30 | 50 |
|---|---|---|---|---|---|---|
| **Layer Level Merge Three Models** | Math | $29.92 \pm 1.53$ | $31.19 \pm 0.36$ | $30.83 \pm 0.28$ | $30.50 \pm 0.54$ | $30.81 \pm 0.48$ |
| | Coding | $16.87 \pm 1.75$ | $15.24 \pm 0.50$ | $15.65 \pm 0.57$ | $15.85 \pm 0.50$ | $15.65 \pm 0.29$ |
| | Translate | $81.78 \pm 0.09$ | $81.67 \pm 0.02$ | $81.72 \pm 0.02$ | $81.68 \pm 0.05$ | $81.80 \pm 0.03$ |
| **Attn/MLP Level Merge Three Models** | Math | $32.17 \pm 0.73$ | $32.78 \pm 0.68$ | $32.85 \pm 0.60$ | $32.02 \pm 0.31$ | $32.68 \pm 0.06$ |
| | Coding | $15.24 \pm 0.86$ | $14.63 \pm 0.01$ | $14.63 \pm 0.50$ | $14.84 \pm 0.57$ | $14.43 \pm 0.29$ |
| | Translate | $81.84 \pm 0.05$ | $81.74 \pm 0.04$ | $81.72 \pm 0.06$ | $81.79 \pm 0.03$ | $81.79 \pm 0.05$ |

$$
f\left(x; \theta_0^i + \sum_{t'=1}^{T} \alpha_{t'}^i \tau_{t'}^i\right) - f(x; \theta_t^i)
$$

$$
= \sum_{t'=1, t' \neq t}^{T} \alpha_{t'}^i \left(f\left(x; \theta_0^i + \tau_{t'}^i\right) - f(x; \theta_0^i)\right) + (\alpha_t^i - 1)\left(f\left(x; \theta_0^i + \tau_t^i\right) - f(x; \theta_0^i)\right) \tag{21}
$$

$$
= \sum_{t'=1, t' \neq t}^{T} \alpha_{t'}^i \Delta f\left(x; \theta_{t'}^i\right) + (\alpha_t^i - 1)\Delta f\left(x; \theta_t^i\right)
$$

where $\Delta f\left(x; \theta_t^i\right) = f\left(x; \theta_t^i\right) - f(x; \theta_0^i)$

Consider $f(x; \theta)$ as a $d$-dimensional vector. We have

$$
\alpha_1^i, \alpha_2^i, \ldots, \alpha_T^i = \arg\min \sum_{t=1}^{T} \mathbb{E}_{x \in H_t^i} \sum_{k=1}^{d} \left(\sum_{t'=1, t' \neq t}^{T} \alpha_{t'}^i \Delta f\left(x; \theta_{t'}^i\right)_k + (\alpha_t^i - 1)\Delta f\left(x; \theta_t^i\right)_k\right)^2 \tag{22}
$$

Taking the derivative of $\alpha_j^i$ and setting the derivative equal to zero, we have

$$
2 \sum_{t=1, t \neq j}^{T} \mathbb{E}_{x \in D_t} \sum_{k=1}^{d} \Delta f\left(x; \theta_j^i\right)_k \left(\sum_{t'=1, t' \neq t}^{T} \alpha_{t'}^i \Delta f\left(x; \theta_{t'}^i\right)_k + (\alpha_t^i - 1)\Delta f\left(x; \theta_t^i\right)_k\right)
$$

$$
+ 2 \mathbb{E}_{x \in D_j} \sum_{k=1}^{d} \Delta f\left(x; \theta_j^i\right)_k \left(\sum_{t'=1, t' \neq j}^{T} \alpha_{t'}^i \Delta f\left(x; \theta_{t'}^i\right)_k + (\alpha_j^i - 1)\Delta f\left(x; \theta_j^i\right)_k\right) = 0 \tag{23}
$$

Let

$$
\boldsymbol{B}_{abc} = \mathbb{E}_{x \in D_a} \sum_{k=1}^{d} \Delta f\left(x; \theta_b^i\right)_k \Delta f\left(x; \theta_c^i\right)_k \tag{24}
$$

We have

$$
\sum_{t=1, t \neq j}^{T} \left(\sum_{t'=1, t' \neq t}^{T} \alpha_{t'}^i \boldsymbol{B}_{tjt'} + (\alpha_t^i - 1)\boldsymbol{B}_{tjt}\right)
$$

$$
+ \left(\sum_{t'=1, t' \neq j}^{T} \alpha_{t'}^i \boldsymbol{B}_{jjt'} + (\alpha_j^i - 1)\boldsymbol{B}_{jjj}\right) = 0 \tag{25}
$$

Then we have

$$\sum_{t=1,t\neq j}^{T}\sum_{t'=1,t'\neq t}^{T}\alpha_{t'}^{i}\boldsymbol{B}_{tjt'} + \sum_{t=1,t\neq j}^{T}\alpha_{t}^{i}\boldsymbol{B}_{tjt} + \sum_{t'=1,t'\neq j}^{T}\alpha_{t'}^{i}\boldsymbol{B}_{jjt'} + \alpha_{j}^{i}\boldsymbol{B}_{jjj} = \sum_{t=1,t\neq j}^{T}\boldsymbol{B}_{tjt} + \boldsymbol{B}_{jjj} \tag{26}$$

We have

$$\sum_{m=1,m\neq j}^{T}(\sum_{t=1,t\neq j}^{T}\boldsymbol{B}_{tjm} - \boldsymbol{B}_{mjm} + \boldsymbol{B}_{mjm} + \boldsymbol{B}_{jjm})\alpha_{m}^{i} + (\sum_{t=1,t\neq j}^{T}\boldsymbol{B}_{tjj} + \boldsymbol{B}_{jjj})\alpha_{j}^{i} = \sum_{t=1}^{T}\boldsymbol{B}_{tjt} \tag{27}$$

Then

$$\sum_{m=1}^{T}(\sum_{t=1}^{T}\boldsymbol{B}_{tjm})\alpha_{m}^{i} = \sum_{t=1}^{T}\boldsymbol{B}_{tjt} \tag{28}$$

Therefore, by differentiating with respect to each $\alpha_{j}^{i}$, where $j \in [1, \ldots, T]$, and setting the derivative equal to zero, we obtain:

$$\sum_{m=1}^{T}(\sum_{t=1}^{T}\boldsymbol{B}_{t0m})\alpha_{m}^{i} = \sum_{t=1}^{T}\boldsymbol{B}_{t0t}$$
$$\sum_{m=1}^{T}(\sum_{t=1}^{T}\boldsymbol{B}_{t1m})\alpha_{m}^{i} = \sum_{t=1}^{T}\boldsymbol{B}_{t1t} \tag{29}$$
$$\cdots$$
$$\sum_{m=1}^{T}(\sum_{t=1}^{T}\boldsymbol{B}_{tTm})\alpha_{m}^{i} = \sum_{t=1}^{T}\boldsymbol{B}_{tTt}$$

We have
$$\boldsymbol{A}[\alpha_{1}^{i}, \alpha_{2}^{i}, ..., \alpha_{T}^{i}]^{\top} = \boldsymbol{b} \tag{30}$$

where

$$\boldsymbol{A}_{j,k} = \sum_{t=1}^{T}\boldsymbol{B}_{tjk}, \boldsymbol{A} \subseteq \mathbb{R}^{T\times T}$$
$$\boldsymbol{b}_{j} = \sum_{t=1}^{T}\boldsymbol{B}_{tjt}, \boldsymbol{b} \subseteq \mathbb{R}^{T} \tag{31}$$

Finally we have
$$[\alpha_{1}^{i}, \alpha_{2}^{i}, ..., \alpha_{T}^{i}]^{\top} = \boldsymbol{A}^{-1}\boldsymbol{b} \tag{32}$$

In the practical implementation, we utilize `np.linalg.solve` to solve Eq.32.

### A.3.2 BALANCE THE INFLUENCE OF EACH SAMPLE

In practical applications of the formula for calculating merging parameters, we observed discrepancies in biases present in data from different tasks. For instance, the average token length varies across tasks (e.g., code data exhibits a greater average length than translation data), and there are differing token distributions (e.g., common token distributions differ between mathematical and code datasets). Consequently, this leads to varying norms of the output feature differences ($\|\Delta f(x;\theta)\|^{2}$) of the models and modules.

To balance the influence of each sample within the formula, we introduce a denominator for each data-related term:

$$\alpha_1^i, \alpha_2^i, \ldots, \alpha_T^i = \arg\min \sum_{t=1}^{T} \mathbb{E}_{x \in D_t} \frac{\left\| f\left(x; \theta_0^i + \sum_{t'=1}^{T} \alpha_{t'}^i \tau_{t'}^i\right) - f(x; \theta_t^i) \right\|^2}{\frac{1}{T} \sum_{t''=1}^{T} \|\Delta f(x; \theta_0^i + \tau_{t''}^i)\|^2} \tag{33}$$

Finally, we obtain that

$$B_{abc} = \mathbb{E}_{x \in D_a} \frac{\sum_{k=1}^{d} \Delta f\left(x; \theta_b^i\right)_k \Delta f\left(x; \theta_c^i\right)_k}{\frac{1}{T} \sum_{t''=1}^{T} \|\Delta f(x; \theta_0^i + \tau_{t''}^i)\|^2} \tag{34}$$

The remaining content is identical to that of the previous section.

### A.3.3 Gather Input Feature Set for Submodule Level Analyzing and Merging

When analyzing the linearity of modules and calculating the merging parameters for module level merging (including layer level, attn/mlp level and head/mlp leve), We should collect the feature set $H_t^i$ corresponding to the input position of each module $i$ first.

In practice, we input the data from dataset $D_t$ into the pre-trained model $\theta_0$ all at once, performing a single forward inference to collect all intermediate features, which serve as the input for each module at the corresponding position.

Especially, in layer level's calculation, the input of $i^{th}$ layer should be the output of the $(i-1)^{th}$ layer in $\theta_0$. In attn/mlp level's calculation, the input of $i^{th}$ attn should be the output of the $(i-1)^{th}$ layer in $\theta_0$, and the input of $i^{th}$ mlp should be the output of the $(i)^{th}$ attn together with output of $(i-1)^{th}$ layer in $\theta_0$. In head level's calculation, the input of head in layer $i^{th}$ should be the output of the $(i-1)^{th}$ layer in $\theta_0$.

### A.3.4 More Details about the Baseline

For Task arithmetic, We explored the hyperparameter merging weights $\alpha \in [0.1, 0.2, \ldots, 0.9, 1.0]$ and selected the best results for reporting in the table.

For DARE, following the recommendations in the paper, we examined various combinations of the hyperparameter dropout probability $drop\_ratio \in [0.6, \ldots, 0.9]$ and merging weights $\alpha \in [0.6, 0.8, 1.0]$, reporting the best results in the table.

The optimal hyperparameters corresponding to the best results in Table 1 and 2 obtained in practice are reported in Table 12.

Table 12: The optimal hyperparameters corresponding to the best results obtained in practice for baslines.

| Hyper-parameters | Math & Code | Math & Translate | Coding & Translate | Math & Coding & Translate |
|---|---|---|---|---|
| Llama-2-7B | | | | |
| DARE | 0.8 1.0 | 0.8 1.0 | 0.8 1.0 | 0.8 1.0 |
| Task Arithmetic | 0.6 | 0.5 | 0.4 | 0.5 |
| Llama-2-13B | | | | |
| DARE | 0.7 1.0 | 0.7 1.0 | 0.8 1.0 | 0.7 1.0 |
| Task Arithmetic | 0.6 | 0.6 | 0.4 | 0.5 |

### A.3.5 More Details at the Head/mlp Level

For the Head/MLP level, due to the architecture of the transformer, the output features corresponding to a single head cannot be directly obtained: The `o_proj` module will gather the intermediate features of each head and map them into a comprehensive attention module output, which can not be directly divided by the arrangement of the heads.

Therefore, we employed certain techniques to achieve this in practice: For each layer, we first collect the input of `o_proj`. Subsequently, we divid `o_proj` into $n\_head$ linear layers according to the parameter relationships associated with each head. The corresponding segments of the collected input are then fed into these corresponding linear layers, producing outputs that serve as the final output features for each head, which will be used for subsequent calculations of the merging parameters.

## A.4   VISUALIZATION RESULTS

Table 13: The actual parameters obtained at the model level. This is one of the results from three random trials.

| Method | Math & Code | Math & Translate | Coding & Translate | Math & Coding & Translate |
|---|---|---|---|---|
| | Llama-2-7B | | | |
| Ours Model Level | [0.851 0.259] | [3.431 -1.537] | [0.237 0.882] | [0.758 0.480 -0.180] |
| | Llama-2-13B | | | |
| Ours Model Level | [1.039 0.132] | [-1.083 1.652] | [-0.463 1.923] | [0.562 0.422 0.062 ] |

## A.5   MORE DISCUSSIONS

### A.5.1   DIFFERENCES BETWEEN THE LINEARLY PROPERTIES STUDIED IN THIS PAPER AND TRADITIONAL LINEAR PROPERTIES

The linearity properties we studied in this paper differ from the Traditional Linear Properties, as illustrated in the following table:

Table 14: Differences between the Linearly Properties Studied in this Paper and Traditional Linear Properties

| | Traditional Linear Properties | Linearity Properties Studied in This Paper |
|---|---|---|
| Definition | Linear relationship between the difference in output features of different input $f_\theta(x) - f_\theta(x_0)$ and the difference in input $x - x_0$. $f_\theta()$ denotes model(module) with parameter $\theta$ | Linear relationship between the difference in output features of the model (module) before and after fine-tuning $f_\theta(x) - f_{\theta_0}(x)$ and the difference in parameters before and after fine-tuning $\theta - \theta_0$. |
| Formula Form | $f_\theta(x) - f_\theta(x_0) \approx C(x - x_0)$ where $C$ is a constant determined by $\theta$. | $f_\theta(x) - f_{\theta_0}(x) \approx C(\theta - \theta_0)$ where $C$ is a constant determined by $\theta_0$ and $x$. |
| Determinants | Determined by the model (module) architecture of $\theta$. | Determined by both the alteration introduced by fine-tuning process and the model (module) architecture of $\theta_0$. |

We also present an example to futher illustrate that there is no particularly intuitive connection between the linearity of the module and the linearity of its submodules:

Consider two contiguous submodules, $\theta_1$ and $\theta_2$. Let $\{\theta_1; \theta_2\}$ denote the composed module.

If $\theta_1$ and $\theta_2$ both satisfy the Traditional Linear Properties, it is obvious that the module $\{\theta_1; \theta_2\}$ which formed by these submodules will also adhere to the Traditional Linear Properties.

However, if $\theta_1$ and $\theta_2$ both satisfy the linearity property studied in this paper, for an arbitrary x, the final output results after applying perturbations $\Delta\theta_1$ and $\Delta\theta_2$ to the two modules is:

| Input | $x$ |
|---|---|
| Output of the first submodule $\theta_1 + \Delta\theta_1$ | $f_{\theta_1}(x) + C_{\theta_1,x}\Delta\theta_1$ |
| Output of the second submodule $\theta_2 + \Delta\theta_2$ | $f_{\theta_2}(f_{\theta_1}(x) + C_{\theta_1,x}\Delta\theta_1) + C_{\theta_2,f_{\theta_1}(x)+C_{\theta_1,x}\Delta\theta_1}\Delta\theta_2$   ...(1) |

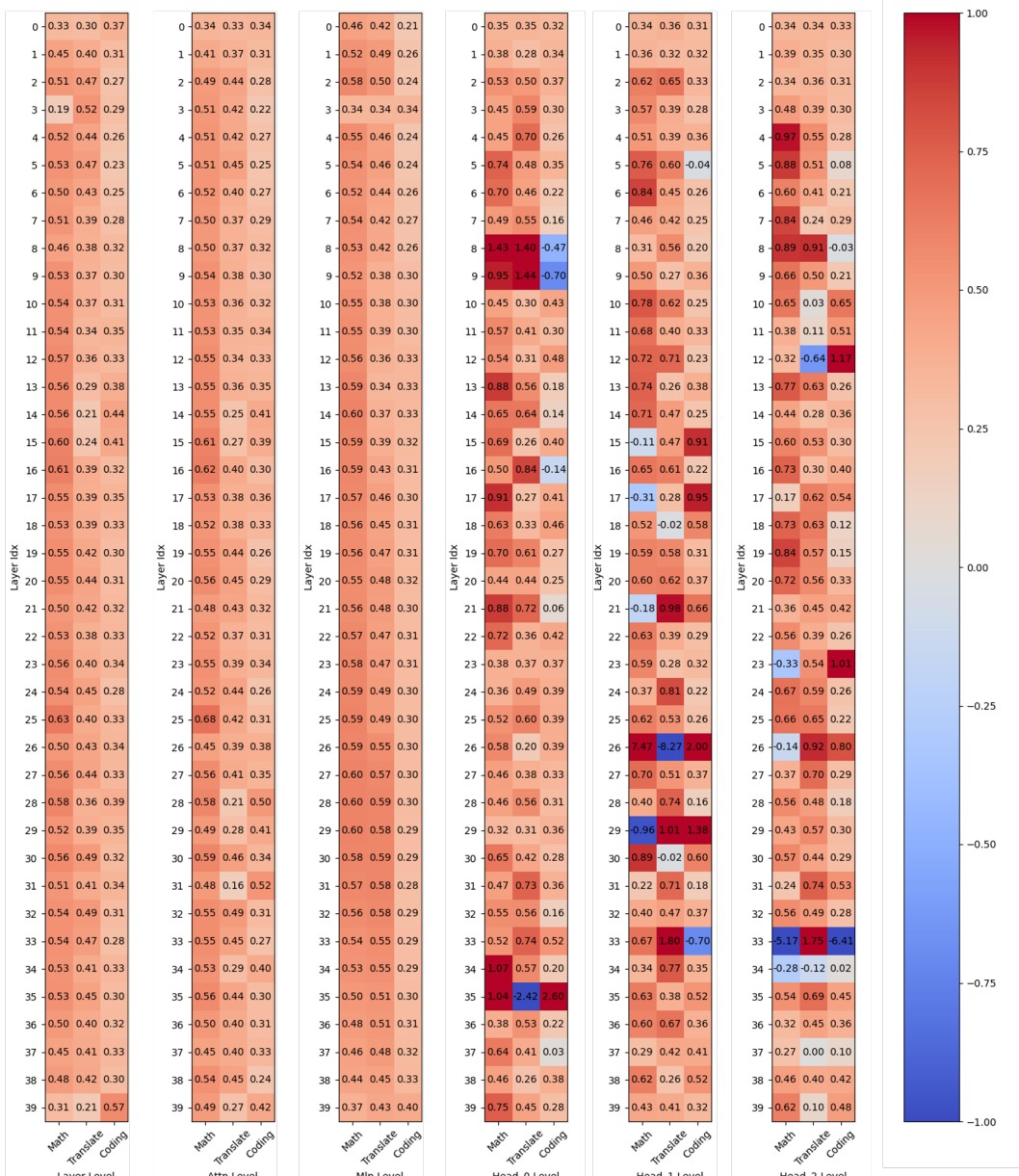

Figure 5: A visual representation of the merging parameters obtained at different levels during the actual merging process is presented. Here, we showcase a result from the merging of three fine-tuned models. Additionally, only a portion of the head's merging parameters is displayed, while the distribution of the merging parameters for the other heads is similar to those presented.

$C_{\theta,x}$ denote a constant determined by $\theta$ and $x$. Now, if $\{\theta_1; \theta_2\}$ as a whole satisfies linearity property studied in this paper, then its output should be $f_{\theta_2}(f_{\theta_1}(x)) + C_{\{\theta_1;\theta_2\},x}\{\Delta\theta_1; \Delta\theta_2\}$ ....(2) when input $x$.

We can observe a significant difference between formulas (1) and (2) , indicating that there is no particularly intuitive connection between the linearity of the module and the linearity of its submodules, which also necessitates further experimental and observational analysis, as what we do in this paper.

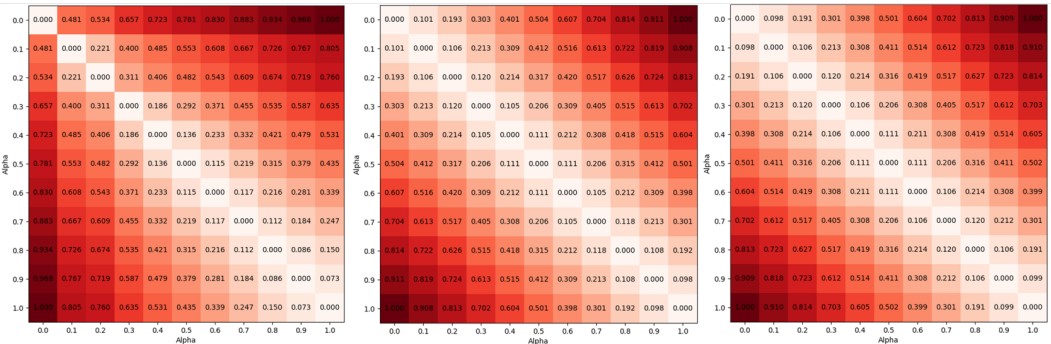

Figure 6: A sample of Visualization of the $\frac{\mathcal{D}(f(x;\theta_i),f(x;\theta_j))}{\mathcal{D}(f(x;\theta),f(x;\theta_0))}$. The image is sourced from the linearity validation of the Llama-2-13B model after fine-tuning for mathematical tasks. From left to right, it displays the model-level linearity validation, the layer-level validation at the 9th layer, and the MLP-level validation at the 9th layer. The latter two validations are already very close to ideal linearity.

### A.5.2 COMPARISON WITH WEMoE AND CAT

Table 15: Comparison with WEMoE and CAT.

| Categories | Ours | BTX(Sukhbaatar et al., 2024)(std MoE) | WEMoE(Tang et al.) | CAT (Prabhakar et al., 2024) |
|---|---|---|---|---|
| Requirements for the Model to be Merged | Same pretrain model, conventional fine-tuning with different task. | Same pretrain model, conventional fine-tuning with different task. | Same pretrain model, conventional fine-tuning with different task. | Same pretrain model, LoRA fine-tuning with different task. |
| Which Part to be Merged | All modules | All modules except MLP | All modules except MLP, The MLP will be merged during inference. | All LoRA's Parameter |
| Hyperparameters or Additional Parameters Required for Merging | Weights for each submodules (such as Layer/Attn/MLP) | Router Network for each MLP | Router Network for each MLP | Weights for each Lora Martix |
| Optimization Methods | Closed-form Solution (Training-free) | Training | Training | Training |
| The Architecture of the Merged Model | Same as Model to be Merged | MoE Architecture (Router for mlp's output Merging Weights) | MoE Architecture (Router for Parameter Merging Weights) | Same as Model to be Merged |

In our approach, merging is performed directly, which may result in different merging outcomes due to the selection of varying data. Our method requires only minimal data and does not necessitate training, yielding a model with a standard architecture. In comparison to our method, WEMoE constructs a MoE architecture model and employs a Router during inference to determine the merging weights for each MLP based on the current input, which also utilizes a linear merging form. As a result, WEMoE may get better merging weights and achieve better results but incurs additional training and inference costs.

### A.5.3 DISCUSSION ON SECOND-ORDER INFORMATION

In our initial setup, considering that the original form of Task Arithmetic is first-order linear, we did not give much consideration to the involvement of second-order information for consistency. As mentioned in the comments, we recognize that further considering the incorporation of second-order information is a very meaningful direction. If we can find methods to utilize second-order information, we may be able to further refine our definitions of linearity and formula expressions, and estimate the linearity of each module with lower error.

Additionally, we might consider iteratively calculating the weight of each module; for example, first calculate the weight of the first layer, then use the fused output features of the first layer to guide the weight calculation of the second layer. This might further exploit the potential connections between modules in different positions. By utilizing second-order information, we might be able to design a new loss function that merges all the weights to be optimized into a simultaneous optimization

process. In future work, we will explore more applications of second-order information in our method.

### A.5.4 Comparison with Git Re Basin, Zipit and MuDSC

Table 16: Comparison with Git Re Basin, Zipit and MuDSC

| Comparison Table | Applicable Module | The Type of Linearity Relied upon during the Merging | Key method for merging |
|---|---|---|---|
| Git Re Basin (Ainsworth et al.) | Linear Layer (full connected layer) in Network | Traditional Linearity Properties | Combinatorial optimization |
| Zipit! (Stoica et al., 2023) | Linear Layer (full connected layer) in Network | Traditional Linearity Properties | Align features by similarity in the activation space |
| MuDSC (Xu et al., 2024b) | Linear Layer (full connected layer) in Network | Traditional Linearity Properties | Align units by similarity in both the parameter space and the activation space |
| Ours | Each submodule level (including layer, attn and MLP) that exhibit Linearity introduced in this paper | Linearity Properties Studied in This Paper | Closed-form solution of optimal merging weights |

As shown in the table, all these methods relies on the Traditional Linearity Property, and can only apply to linear layers. In summary, we believe that our method has several novel contributions that make it distinct from Git Re Basin (Ainsworth et al.), Zipit!(Stoica et al., 2023) and MuDSC(Xu et al., 2024b).

### A.5.5 Discussion on Finely-grained Components

Further decomposing the model into more finely-grained components is indeed an intuitive and promising direction for exploration. In the paper, we conducted some related experiments. Based on the structure of the LLM model, we further divided the attention layer into individual heads, independently estimating their linearity and ultimately testing their performance, but revealing unstable outcomes at this level (as shown in Table 3).

Here we propose a potential explanation for the occurrence of this situation. Numerous studies (Wang et al., 2023; Chen et al., 2024; Zhang et al., 2024) suggest that each head in the model performs different roles during inference. Following fine-tuning across various tasks, certain heads may adopt distinct functions. However, our objective function may necessitate a single head to simultaneously average among multiple heads with diverse functionalities. This requirement can be quite challenging for an individual head and may ultimately result in a collapse of functionality in the merged head.

However, this may not imply that the linearity of the head or more finely-grained components is unhelpful for the merging process. As we mentioned above, the relationship between the linearity of the module and the linearity of its submodules is not straightforward; therefore, further experiments and analyses are required. We will further explore whether there are better ways to address the aforementioned issues in future work. For instance, based on the architecture of the LLM, we could further decompose each module into individual linear layers (decompose attention into qkvo projection linear layers, and MLP into two linear layers). Alternatively, we could incorporate interpretability methods(Wang et al., 2023) to improve understanding and identify which head is most relevant to a specific task.

### A.5.6 Discussion about Activation Functions

In this paper, we did not make any changes to the architecture of any part of the model, and all nonlinear activation functions remained unaltered. Regarding the achievement of such linearity without modifying the activation functions, we believe this is due to the many differences between the Traditional Linear Properties and the Linearity Properties Studied in This Paper discussed above.

The connection between nonlinear activation functions and Traditional Linear Properties has been widely studied, but the connection between nonlinear activation functions and the Linearity Properties Studied in This Paper still awaits exploration. We believe that activation functions are likely to play a crucial role in the emergence of linearity.

Currently, the vast majority of LLMs (including Qwen and Llama-2) use the SwiGLU activation function. Compared to other activation functions such as ReLU and GELU, SwiGLU is more complex and nonlinear, yet it also enhances performance. We hypothesize that replacing SwiGLU with more linear activation functions could improve overall linearity, which would be beneficial for our method. However, we must carefully consider the balance between the immediate performance drop caused by the replacement and the performance gains achieved through the merging, which requires further experiments and analysis.

However, it is difficult to find other identical LLMs, differing only in their activation function, for fair comparison due to the enormous cost of training a LLM. We plan to conduct experiments related to activation functions in a controlled environment as described in Allen-Zhu & Li (2023) in future work.

### A.5.7 DISCUSSION ABOUT HOW TO LEVERAGE LINEARITY AT THE HEAD LEVEL

1. Our initial proposal involves collaborating with interpretability methods Wang et al. (2023). Initially, we can use interpretability methods to determine which head is actually more relevant to a particular task. By obtaining weights through relevance, we can further adjust the originally explored weights to make full use of the capability of each head.

2. The second proposal involves modifying our loss function: our original loss is quite simple and only explores the output feature level. We can refer to other approaches (Stoica et al., 2023; Xu et al., 2024b) to also explore parameter space, making the explored weights more reasonable and enhancing the use of linearity.

3. The third proposal is to use our linearity to analyze which task a given head is actually more related to (since our linearity is introduced by fine-tuning). This might potentially become a new method of interpretability.

### A.5.8 DISCUSSION ABOUT THE EFFECTS OF MODEL DEPTH

For a fair and convincing comparison, we utilized existing backbones like Qwen and Llama to examine how different depths impact linearity, using the Non-linearity Score as an indicator, which is introduced in our paper and with smaller score indicating closer adherence to ideal linearity.

Table 17: Non-linearity Score for Different Models

| Level | Qwen-2-0.5B | Qwen-2.5-7B | Llama-2-7B | Llama-2-13B |
|-------|-------------|-------------|------------|-------------|
| Model | $2.1130 \pm 0.5250$ | $4.2559 \pm 0.7037$ | $2.7637 \pm 0.3442$ | $2.1929 \pm 0.2271$ |
| Layer | $0.4430 \pm 0.7331$ | $0.1412 \pm 0.3019$ | $0.1645 \pm 0.4329$ | $0.1675 \pm 0.3760$ |
| Attn | $0.2439 \pm 0.1631$ | $0.0649 \pm 0.0592$ | $0.0229 \pm 0.0107$ | $0.0250 \pm 0.0115$ |
| MLP | $0.3112 \pm 0.8353$ | $0.0810 \pm 0.2068$ | $0.2030 \pm 0.5655$ | $0.1622 \pm 0.4295$ |

From the table, we can observe:

1. At the model level, linearity does not seem to have a clear relationship with model depth. For the Qwen model, the linearity of the 0.5B model is better than the deeper 7B model, while for Llama, linearity is greater in the 7B model compared to the deeper 13B.

2. At the Layer/Attn/MLP level, within the transformer architecture of decoder language models, the depth of each Layer/Attn/MLP across different model sizes can be regarded as the same, with variations occurring in the width. Overall, wider modules tend to exhibit slightly better linearity, which aligns with conclusions in NTK-related works such as Jacot et al. (2018) and Chizat et al. (2019).

### A.5.9 DISCUSSION ABOUT THE PHENOMENON THAT QWEN-2.5-7B HAS MUCH HIGHER NON-LINEARITY SCORE THAN OTHERS

In order to provide a possible explanation for the phenomenon that the non-linearity score for Qwen-2.5-7B is much higher than others, we conducted further experiments based on Property 1 from the paper.

Specifically, we first computed the parameter difference before and after fine-tuning for each model, denoted as $\tau = \theta - \theta_0$. We then used a parameter $\alpha$ to weight this parameter difference and added it back to the pre-trained model $\theta_0$ to obtain the interpolated model $\theta_0 + \alpha\tau$. The L2 distance between the features output by this model $\theta_0 + \alpha\tau$ and the features output by the fine-tuned model $\theta$ was computed and is presented in the table below. (When $\alpha = 0$, the interpolated model becomes the pre-trained model $\theta_0$. When $\alpha = 1$, the interpolated model becomes the fine-tuned model $\theta$, the distance will be 0. All distances are normalized. The Non-linearity Score is directly proportional to the difference between the value for the specific model and the value for ideal linearity in the table.)

Table 18: Model Linearity and Corresponding Non-linearity Score

| Model | $\alpha = 0.0$ | $\alpha = 0.1$ | $\alpha = 0.2$ | $\alpha = 0.3$ | $\alpha = 0.4$ | $\alpha = 0.5$ | $\alpha = 0.6$ | $\alpha = 0.7$ | $\alpha = 0.8$ | $\alpha = 0.9$ | $\alpha = 1.0$ | Non-linearity Score |
|---|---|---|---|---|---|---|---|---|---|---|---|---|
| Ideal Linearity | 1.000 | 0.900 | 0.800 | 0.700 | 0.600 | 0.500 | 0.400 | 0.300 | 0.200 | 0.100 | 0.000 | 0 |
| Qwen-2.5-7B | 1.000 | 0.911 | 0.879 | 0.841 | 0.805 | **0.731** | **0.429** | 0.277 | 0.169 | 0.085 | 0.000 | 4.2559 |
| Qwen-2-0.5B | 1.000 | 0.923 | 0.828 | 0.763 | 0.719 | 0.655 | 0.570 | 0.427 | 0.262 | 0.125 | 0.000 | 2.1130 |
| Llama-2-7B | 1.000 | 0.871 | 0.726 | 0.637 | 0.545 | 0.483 | 0.370 | 0.263 | 0.169 | 0.080 | 0.000 | 2.7637 |
| Llama-2-13B | 1.000 | 0.805 | 0.760 | 0.635 | 0.531 | 0.435 | 0.339 | 0.247 | 0.150 | 0.073 | 0.000 | 2.1929 |

For a model that satisfies linearity, the distance between the interpolated model's output features and the fine-tuned model's output features should decrease linearly as $\alpha$ increases (as shown in the first row of the table). However, we observe from the table that the Qwen-2.5-7B model experiences a very steep decline between $\alpha = 0.5$ and $\alpha = 0.6$, which does not occur with other backbones. Such a steep decline does not satisfy linearity well, resulting in the non-linearity score for Qwen-2.5-7B in model level being much higher than others.

This is a very interesting phenomenon. We find that the performance of Qwen-2.5-7B is very similar to a sigmoid function: when the parameter difference (or $\alpha$) is far from the threshold, the change in its output features is relatively slow, whereas the change is rapid when the parameter difference (or $\alpha$) approaches the threshold.

We suspect that this phenomenon is related to the model's foundational capabilities, as Qwen-2.5-7B is a model with significantly stronger performance in various aspects compared to other models (according to official website data). To explore the specific reasons for this intriguing phenomenon, we will conduct more experiments and analyses in our future work.

