# OpenReview forum: "Leveraging Submodule Linearity Enhances Task Arithmetic Performance in LLMs"
_ICLR.cc/2025/Conference — ICLR 2025 Poster_

### Official Review · Reviewer_KQV8 · 2024-10-22

**Soundness:** 2
**Presentation:** 2
**Contribution:** 2
**Rating:** 6
**Confidence:** 3

**Summary:**

This work is inspired by models demonstrating linearity that can improve the performance of task arithmetic, and hypothesize that linearity already exists within the model’s submodules. Linearity refers to the linear relationship between the differences in model weights and the differences in output features caused by fine-tuning. A model framework that independently merges these submodules is proposed to enhance task arithmetic and hence lead to superior multi-task models. The author justify their motivation by comparing the non-linearity scores between the full model and the submodules. Using submodules instead full models requires computation of the optimal merging weights after splitting the models into submodules that exhibit linear characteristics. Experiments are conducted on fine-tuned models based on Llama-2 and the results indicate superior merging performance with task arithmetic.

**Strengths:**

The paper propses an interesting idea of decomposing the models into submodules that exhibit linear characteristics. The non-linearity score is proposed and merging the weights this way is novel. In particular, Figure 1 and Figure 3 clearly illustrate the motivation behind this paper and the weight merging is well-defined. The results and the ablation study support the author's claim well. The contribution of Layer level and Attention/MLP level weight merging is convincing.

**Weaknesses:**

One relevant topic concerning linearity is the non-linear activation function. This paper focused on the submodules and weight merging, but lacks a thorough discussion on activation functions. Figures show that the non-linearity score for all different submodeules are near zero. Why is that? Have you discarded all non-linear activation functions in submodules? If that is not the case, how do they affect the linearity assumptions and why do they not contribute to a larger non-linerarity score? Please include a discussion of activation functions.

**Questions:**

1. Merging weights in head level appears to give unstable outcomes in Table 3, and the hypothesis here is that the representation functionalities can be averaged out. Is there any potential method to avoid that?

2. The fine-tuned models are based on Llama-2, and the 7B model tends to show stronger non-linearity than the 13B model. Have you considered the influence of the varying depth of the model and data size, and the implications with repect to universal approximation theorem?

3. Is there any family of activation function more suitable for your model framework than others (e.g. ReLU, GELU, Swish) and what is their impact on the linearity of submodules and overall merging performance?

---

> ### Author Response · Authors · 2024-11-19
> **Response by Authors**
>
> ``Part 1/2``
>
> ---
>
>
>
>
> **Response by Authors:**
>
> Thank you for taking the time to review our work and provide valuable comments! We appreciate your comments and suggestions, which have helped us improve our paper. Please find below our detailed responses to your comments.
>
> ---
> **Weaknesses and Questions 3 :**
> > One relevant topic concerning linearity is the non-linear activation function. This paper focused on the submodules and weight merging, but lacks a thorough discussion on activation functions. Figures show that the non-linearity score for all different submodeules are near zero. Why is that? Have you discarded all non-linear activation functions in submodules? If that is not the case, how do they affect the linearity assumptions and why do they not contribute to a larger non-linerarity score? Please include a discussion of activation functions.
>
> >  Is there any family of activation function more suitable for your model framework than others (e.g. ReLU, GELU, Swish) and what is their impact on the linearity of submodules and overall merging performance?
>
> **Response to Weaknesses and Questions 3:**
>
> We are very grateful to you for bringing the new perspective of activation functions to us. We will add the discussion below about activation functions into our revision.
>
> In this paper, we did not make any changes to the architecture of any part of the model, and all nonlinear activation functions remained unaltered. Regarding the achievement of such linearity without modifying the activation functions, we believe this is due to the many differences between the Traditional Linear Properties and the Linearity Properties Studied in This Paper. We illustrate these differences in a table:
>
> |  | Traditional Linear Properties | Linearity Properties Studied in This Paper |
> |-------|-------|-------|
> | Definition | Linear relationship between the difference in output features of different input $f_\theta(x)-f_\theta(x_0)$ and the difference in input $x-x_0$. $f_{\theta}()$denotes model(module) with paramter $\theta$ | Linear relationship between the difference in output features of the model (module) before and after fine-tuning $f_{\theta}(x) - f_{\theta_0}(x)$ and the difference in parameters before and after fine-tuning $\theta - \theta_0$. |
> | Formula Form | $f_\theta(x)-f_\theta(x_0) \approx C(x-x_0)$ where $C$ is a constant determined by $\theta$. | $f_{\theta}(x) - f_{\theta_0}(x) \approx C(\theta - \theta_0)$ where $C$ is a constant determined by $\theta_0$ and $x$. |
> | Determinants | Determined by the model (module) architecture of $\theta$. | Determined by both the alteration introduced by fine-tuning process and the model (module) architecture of $\theta_0$. |
>
>
> We are very appreciative of the new perspective on activation functions that you have brought to our attention. The connection between nonlinear activation functions and Traditional Linear Properties has been widely studied, but the connection between nonlinear activation functions and the Linearity Properties Studied in This Paper still awaits exploration. We believe that activation functions are likely to play a crucial role in the emergence of linearity.
>
> Currently, the vast majority of LLMs (including Qwen and Llama-2) use the SwiGLU activation function. Compared to other activation functions such as ReLU and GELU, SwiGLU is more complex and nonlinear, yet it also enhances performance. We hypothesize that replacing SwiGLU with more linear activation functions could improve overall linearity, which would be beneficial for our method. However, we must carefully consider the balance between the immediate performance drop caused by the replacement and the performance gains achieved through the merging, which requires further experiments and analysis.
>
> However,it is difficult to find other identical LLMs, differing only in their activation function, for fair comparison due to the enormous cost of training a LLM. We plan to conduct experiments related to activation functions in a controlled environment as described in [1] in future work.
>
> ---

---

> > ### Author Response · Authors · 2024-11-19
> > **Response by Authors**
> >
> > ``Part 2/2``
> >
> > ---
> >
> >
> >
> > **Questions 1 :**
> > >  Merging weights in head level appears to give unstable outcomes in Table 3, and the hypothesis here is that the representation functionalities can be averaged out. Is there any potential method to avoid that?
> >
> >
> > **Response to Questions 1 :**
> >
> > Thank you very much for your feedback. Following your valuable comments, we have discussed how to leverage linearity at the head level and potentially at more granular levels. We will incorporate this discussion into our revised version.
> >
> > 1. Our initial proposal involves collaborating with interpretability methods [1, 2]. Initially, we can use interpretability methods to determine which head is actually more relevant to a particular task. By obtaining weights through relevance, we can further adjust the originally explored weights to make full use of the capability of each head.
> >
> > 2. The second proposal involves modifying our loss function: our original loss is quite simple and only explores the output feature level. We can refer to other approaches [3, 4] to also explore parameter space, making the explored weights more reasonable and enhancing the use of linearity.
> >
> > 3. The third proposal is to use our linearity to analyze which task a given head is actually more related to (since our linearity is introduced by fine-tuning). This might potentially become a new method of interpretability.
> >
> > Thanks for your feedback again.
> >
> >
> >
> > ---
> > **Questions 2 :**
> > > The fine-tuned models are based on Llama-2, and the 7B model tends to show stronger non-linearity than the 13B model. Have you considered the influence of the varying depth of the model and data size, and the implications with repect to universal approximation theorem?
> >
> > **Response to Questions 2 :**
> >
> > Thank you very much for your suggestions. Following your advice, we conducted ablation experiments to investigate the effects of model depth and data quantity on linearity and our method. We will include this discussion in our revision.
> >
> > **Impact of Model Depth on Linearity**
> >
> > For a fair and convincing comparison, we utilized existing backbones like Qwen and Llama to examine how different depths impact linearity, using the Non-linearity Score as an indicator, which is introduced in our paper and with smaller score indicating closer adherence to ideal linearity.
> >
> > **Table: Non-linearity Score for Different Models**
> >
> > |Level|Qwen-2-0.5B|Qwen-2.5-7B|Llama-2-7B|Llama-2-13B|
> > |-|-|-|-|-|
> > |Model|2.1130 +- 0.5250 |4.2559 +- 0.7037|2.7637 +- 0.3442 |2.1929 +- 0.2271|
> > |Layer|0.4430 +- 0.7331|0.1412 +- 0.3019|0.1645 +- 0.4329 |0.1675 +- 0.3760|
> > |Attn|0.2439 +- 0.1631|0.0649 +- 0.0592 |0.0229 +- 0.0107 |0.0250 +- 0.0115|
> > |MLP|0.3112 +- 0.8353|0.0810 +- 0.2068 |0.2030 +- 0.5655 |0.1622 +- 0.4295|
> >
> > From the table, we can observe:
> >
> > 1. At the model level, linearity does not seem to have a clear relationship with model depth. For the Qwen model, the linearity of the 0.5B model is better than the deeper 7B model, while for Llama, linearity is greater in the 7B model compared to the deeper 13B.
> >
> > 2. At the Layer/Attn/MLP level, within the transformer architecture of decoder language models, the depth of each Layer/Attn/MLP across different model sizes can be regarded as the same, with variations occurring in the width. Overall, wider modules tend to exhibit slightly better linearity, which aligns with conclusions in NTK-related works such as [4].
> >
> > **Impact of Data Quantity**
> >
> > We conducted an ablation study about the amount of data used in our method.
> >
> > **Table: Ablation on the Num of Data used**
> >
> > | Data Num per Task | 1 | 3 |10 | 30 | 50|
> > |-|-|-|-|-|-|
> > | **Layer Level Merge Three Models** | | | | | |
> > | Math|29.92 ± 1.53 |31.19 ± 0.36 | 30.83 ± 0.28| 30.50 ± 0.54 | 30.81 ± 0.48|
> > | Coding |16.87 ± 1.75 |15.24 ± 0.50 | 15.65 ± 0.57| 15.85 ± 0.50 | 15.65 ± 0.29|
> > | Translate | 81.78 ± 0.09 | 81.67 ± 0.02| 81.72 ± 0.02| 81.68 ± 0.05 | 81.80 ± 0.03|
> > |**Attn/MLP Level Merge Three Models** | | | | | |
> > | Math |32.17 ± 0.73| 32.78 ± 0.68| 32.85 ± 0.60| 32.02 ± 0.31| 32.68 ± 0.06|
> > | Coding | 15.24 ± 0.86| 14.63 ± 0.01| 14.63 ± 0.50| 14.84 ± 0.57| 14.43 ± 0.29|
> > | Translate | 81.84 ± 0.05| 81.74 ± 0.04| 81.72 ± 0.06| 81.79 ± 0.03| 81.79 ± 0.05|
> >
> > From the table, we can see that our method shows consistent performance across different data quantities, sometimes achieving good results with as few as 3 data per task.
> >
> >
> >
> > ---
> >
> > **References**
> >
> > [1] Zeyuan Allen-Zhu, Yuanzhi Li.  Physics of Language Models
> >
> > [2] Meng, Kevin, et al. Locating and editing factual associations in GPT.  NIPS 2022
> >
> > [3] Zhang, Fred, and Neel Nanda. Towards best practices of activation patching in language models: Metrics and methods. ICLR 2024
> >
> > [4] Arthur Jacot, et al. Neural tangent kernel: Convergence and generalization in neural networks. NIPS 2018
> >
> >
> >
> > ---
> >
> > We hope that our response has addressed your concerns. If you have any further questions or suggestions, please do not hesitate to let us know. Thank you once again for your valuable feedback!

---

> ### Comment · Reviewer_KQV8 · 2024-11-20
> **Qwen-2.5-7B**
>
> I thank the author for answering my questions. The new results and discussion are satisfying. Meanwhile, the non-linearity score for Qwen-2.5-7B in the new results seems to be much higher than others, not just Qwen-2-0.5B. Could you give a possible explanation to this?

---

> > ### Author Response · Authors · 2024-11-20
> > **Response to Reviewer KQV8's Comment**
> >
> > Thank you very much for your appreciation and feedback! We are very pleased to answer your questions.
> >
> > In order to provide a possible explanation for the phenomenon that the non-linearity score for Qwen-2.5-7B is much higher than others, we conducted further experiments based on Property 1 from the paper.
> >
> > Specifically, we first computed the parameter difference before and after fine-tuning for each model, denoted as $\tau = \theta - \theta_0$. We then used a parameter $\alpha$ to weight this parameter difference and added it back to the pre-trained model $\theta_0$ to obtain the interpolated model $\theta_0 + \alpha\tau$. The L2 distance between the features output by this model $\theta_0 + \alpha\tau$ and the features output by the fine-tuned model $\theta$ was computed and is presented in the table below. (When $\alpha=0$, the interpolated model becomes the pre-trained model $\theta_0$. When $\alpha=1$, the interpolated model becomes the fine-tuned model $\theta$, the distance will be 0. All distances are normalized. The Non-linearity Score is directly proportional to the difference between the value for the specific model and the value for ideal linearity in the table.)
> >
> >
> > | Model        | $\alpha=0.0$ | $\alpha=0.1$ | $\alpha=0.2$ | $\alpha=0.3$ | $\alpha=0.4$ | $\alpha=0.5$ | $\alpha=0.6$ | $\alpha=0.7$ | $\alpha=0.8$ | $\alpha=0.9$ | $\alpha=1.0$ | Corresponding Non-linearity Score |
> > |---------------|----------------|----------------|----------------|----------------|----------------|----------------|----------------|----------------|----------------|----------------|----------------|-------------------|
> > | ideal linearity | 1.000          | 0.900          | 0.800          | 0.700          | 0.600          | 0.500          | 0.400          | 0.300          | 0.200          | 0.100          | 0.000          | 0            |
> > | Qwen-2.5-7B   | 1.000          | 0.911          | 0.879          | 0.841          | 0.805          | **0.731**          | **0.429**          | 0.277          | 0.169          | 0.085          | 0.000          | 4.2559            |
> > | Qwen-2-0.5B   | 1.000          | 0.923          | 0.828          | 0.763          | 0.719          | 0.655          | 0.570          | 0.427          | 0.262          | 0.125          | 0.000          | 2.1130           |
> > | Llama-2-7B    | 1.000          | 0.871          | 0.726          | 0.637          | 0.545          | 0.483          | 0.370          | 0.263          | 0.169          | 0.080          | 0.000          | 2.7637           |
> > | Llama-2-13B   | 1.000          | 0.805          | 0.760          | 0.635          | 0.531          | 0.435          | 0.339          | 0.247          | 0.150          | 0.073          | 0.000          | 2.1929           |
> >
> >
> >
> >
> > For a model that satisfies ideal linearity, the distance between the interpolated model's output features and the fine-tuned model's output features should decrease linearly as $\alpha$ increases (as shown in the first row of the table). However, we observe from the table that the Qwen-2.5-7B model experiences a very steep decline between $\alpha=0.5$ and $\alpha=0.6$, which does not occur with other backbones. Such a steep decline does not satisfy linearity well, resulting in the non-linearity score for Qwen-2.5-7B in model level being much higher than others.
> >
> > This is a very interesting phenomenon. We find that the changes of Qwen-2.5-7B's output feature is very similar to a sigmoid function: when the parameter difference (or $\alpha$) is far from the threshold, the change in its output features is relatively slow, whereas the change is rapid when the parameter difference (or $\alpha$) approaches the threshold.
> >
> > We suspect that this phenomenon is related to the model's foundational capabilities, as Qwen-2.5-7B is a model with significantly stronger performance in various aspects compared to other models (according to official website). To explore the specific reasons for this intriguing phenomenon, we will conduct more experiments and analyses in our future work.
> >
> > Thank you very much for your feedback and for helping us to discover such an interesting phenomenon!

---

> > > ### Comment · Reviewer_KQV8 · 2024-11-22
> > > **Thank you**
> > >
> > > Thank you for providing this insightful explanation. I have raised my score to 6.

---

> ### Author Response · Authors · 2024-11-22
>
> Dear Reviewer KQV8,
>
> Glad to hear that your concerns are addressed well! Thank you for raising the score.
>
> Thank you once again for taking the time to review our paper!
>
> Best regards,
>
> Authors of # 3017

---

### Official Review · Reviewer_g3he · 2024-11-03

**Soundness:** 2
**Presentation:** 3
**Contribution:** 2
**Rating:** 6
**Confidence:** 4

**Summary:**

The paper tackles task vector-based model merging, following a line of approaches that obtain a multi-task model by adding to a pretrained model the differences between fine-tuned models and their pretrained base.  In particular, the work builds on top of previous work relating the effectiveness of task vectors to a particular kind of linearity such that if a model is taken on the line connecting pretrained and fine-tuned, its output features should be on the line connecting the features of the pretrained and those of the fine-tuned. The treatment starts by defining a non-linearity score that measures how much this linearity property is violated, showing this score to be much lower for individual layers than for the whole model. The authors then leverage this finding and propose merging individual layers, deriving a closed form for the layer weights based on the linearity requirement.  Experiments show the approach to be slightly beneficial for two LLAMA-based architectures when merging a model with math, coding, and translating capabilities.

**Strengths:**

- The paper is generally clear and easy to read. The experiments make intuitive sense and are well explained, with enough details to be reproduced.
- The layer-wise linearity perspective is novel. Overall, I find layer-wise approaches to better leverage the structure of the network when compared with standard vanilla task vectors. The proposed non-linearity score is novel, well-motivated, and intuitive.

**Weaknesses:**

- First and foremost, I am not convinced that the results warrant the added complexity: overall, the performance of the approach is in the same ballpark as vanilla task arithmetic while not inheriting its intuitiveness and simplicity. While it exhibits some slight improvements in a portion of the (limited) evaluation settings, these are not substantial nor consistent enough.
- Still on the evaluation, the paper only considers two LLAMA-based architectures and only a subset of the evaluation settings considered by the previous works. For instance, Task Arithmetics also considers the GLUE benchmark and a suite of computer vision benchmarks, while DARE considers GLUE and OpenLLM. An exhaustive evaluation is necessary to rule out possibly lucky choices of models and datasets that do not guarantee the effectiveness of the approach in general.
- The baselines are also somewhat limited, some possible works to consider are *e.g.* [1], [2] and [3].
- The main finding of the paper, that is the fact that modules show more linearity than the whole model, obtained as a composition of all those intermediate modules, is not surprising. In fact, if the intermediate modules have non-zero non-linearity scores, the overall non-linearity score will grow as these ones accumulate.
- Experiments have shown that merging three models exhibits a lower projection distance than merging just two, highlighting the potential of merging even more models. However, the experiments only merge up to three models. Can the trend be confirmed? More experiments would help clarify this aspect.

[1] Yadav, Prateek, et al. "Ties-merging: Resolving interference when merging models." *Advances in Neural Information Processing Systems* 36 (2024).

[2] Davari, MohammadReza, and Eugene Belilovsky. "Model breadcrumbs: Scaling multi-task model merging with sparse masks." ECCV 2024.

[3] Wang, Ke, et al. "Localizing Task Information for Improved Model Merging and Compression." *Forty-first International Conference on Machine Learning*.

**Questions:**

- The requirement of a small set of data per task is reasonable in practice, but what if no data is available? Can you still do something?
- Maybe add a few more compatible tasks so that merging more than three models reveals some interesting trends as the number of tasks increases. Maybe your framework works even better compared to the baselines when the number of tasks is raised? Or perhaps the advantage becomes less pronounced?
- Minor suggestion: In Property 2, I would really avoid using footnotes over equations as these look like powers instead.

---

> ### Author Response · Authors · 2024-11-19
> **Response by Authors**
>
> ``Part 1/4``
>
> ---
>
>
>
> **Response by Authors**
>
> Thank you for reviewing our paper and providing valuable comments. We would like to acknowledge the time and effort you have invested in reviewing our work. In response to your comments, we have provided a detailed response below.
>
> ---
> **Weaknesses 1:**
> > First and foremost, I am not convinced that the results warrant the added complexity: overall, the performance of the approach is in the same ballpark as vanilla task arithmetic while not inheriting its intuitiveness and simplicity. While it exhibits some slight improvements in a portion of the (limited) evaluation settings, these are not substantial nor consistent enough.
>
> **Response to Weaknesses 1:**
> Thank you for your valuable suggestion and feedback. We have conducted some additional experiments to further demonstrate the efficiency of our method.
>
> Firstly, to demonstrate the advantages of our method in scenarios where more models are merged simultaneously, we start with the fail case of code & translate in LLaMA-2-7B mentioned by the reviewer F2K5. We increase the number of task-specific models merged simultaneously gradually and calculate the average performance of the merged models.
>
> Table :
> | Avg. Acc                  | Coding & Translate | Coding & Translate & Cosmos_qa | Coding & Translate & Cosmos_qa & Ropes | Coding & Translate & Cosmos_qa & Ropes & Winogrande |
> |----|---|---------|-------|---------|
> | Task Arithmetic           | 0.5111    | 0.5643      | 0.5257    | 0.4666      |
> | Ours   | 0.4980   | 0.5581  | 0.5382      | 0.5489      |
> | Gap $\Delta $              | -2.6\%             | -1.1\%   | +2.3\%    | +14.9\%        |
>
> In the table, we can observe that as the number of models increases, our method gradually surpasses Task Arithmetic and the performance gap becomes larger, which demonstrates the efficiency of our method particularly when more models are merged simultaneously.
>
> We also have conducted additional experiments utilizing the new datasets. The results are presented below, demonstrating that our methods consistently exhibit impressive performance.
>
> **Table: Experimental results on LLaMA-2-7B with More Dataset: Winogrande, Cosmos qa and Ropes**
> | Merging Two Models               | Winogrande | Cosmos qa | Avg   | &&& | Winogrande | Ropes  | Avg   | &&& | Cosmos qa | Ropes  | Avg   |
> |-----|------------|-----------|-------|-----|------------|--------|-------|-----|-----------|--------|-------|
> | Fine-tuned Model                 | 0.8226     | 0.8472    | 0.8349| &&& | 0.8226     | 0.7542 | 0.7884| &&& | 0.8472    | 0.7542 | 0.8007|
> | Task Arithmetic                  | 0.7222     | 0.7185    | 0.7203| &&& | 0.7214     | 0.5428 | 0.6321| &&& | 0.7348    | 0.5875 | 0.6611|
> | Ours Attn/MLP Level              | 0.6842     | 0.8418    | **0.7630**| &&& | 0.6947     | 0.6998 | **0.6972**| &&& | 0.7307    | 0.6859 | **0.7083**|
>
> | Merging Three Models             | Winogrande | Cosmos qa | Ropes  | Avg   |
> |------|------------|-----------|--------|-------|
> | Fine-tuned Model                 | 0.8226     | 0.8472    | 0.7542 | 0.808 |
> | Task Arithmetic                  | 0.7125     | 0.6413    | 0.5585 | 0.6374|
> | Ours Attn/MLP Level              | 0.7149     | 0.6870    | 0.5670 | **0.6563**|
>
>
> We also provide potential explanations for the few fail cases mentioned by the reviewer F2K5 (such as coding & translation in LLaMA-2-7B). Our loss design is intended to ensure that the output features of the merged submodule for data from different tasks closely align with the output features of the corresponding task-specific models. To visualize this, we employ t-SNE to observe the distribution of features output by each module when different data inputs are provided. (Please refer to this anonymous link https://anonymous.4open.science/r/rebuttal-tsne-result-select for the samples of t-SNE result, as the OpenReview website does not support direct image uploads.)
>
> We found that the output feature distributions for code data and translate data on various modules in LLaMA-2-7B tend to be relatively close overall (see table below), and sometimes even was interwoven (see the t-SNE result in the link), which contradicts the goal of our loss design and may have led to this fail case. These fail cases occur in only a few situations, with over 90% of the results demonstrating the efficiency of our method.
>
> **Table : Avg. Norm Distance of Output Feature in all t-SNE results**
> |        | Avg. Norm Distance of Output Feature |
> |-----|------|
> | Math vs. Coding         | 0.3475     |
> | Math vs. Translate      | 0.3758    |
> | Coding vs. Translate    | 0.2765      |
>
> In our future work, we will focus on improving these rare fail cases. We plan to explore more efficient methods that utilize linearity, such as incorporating second-order information as suggested by reviewer SeC4, in order to further enhance the overall performance of our approach.
>
> ---

---

> > ### Author Response · Authors · 2024-11-19
> > **Response by Authors**
> >
> > ``Part 2/4``
> >
> > ---
> >
> >
> >
> > **Weaknesses 2:**
> > > Still on the evaluation, the paper only considers two LLAMA-based architectures and only a subset of the evaluation settings considered by the previous works. For instance, Task Arithmetics also considers the GLUE benchmark and a suite of computer vision benchmarks, while DARE considers GLUE and OpenLLM. An exhaustive evaluation is necessary to rule out possibly lucky choices of models and datasets that do not guarantee the effectiveness of the approach in general.
> >
> > **Response to Weaknesses 2:**
> > Thank you for your constructive comments. We appreciate your concern about the overall effectiveness of the approach. In response to your valuable feedback, we have conducted additional experiments utilizing the new backbones Qwen-2-0.5B and Qwen-2.5-7B, as well as new datasets such as Winogrande (from OpenLLM), Cosmos_qa, and Ropes. The results are presented below, demonstrating that our methods consistently exhibit impressive performance. The settings of training and merging are the same as in the paper. The experimental results on LLaMA-2-7B with More Dataset including Winogrande, Cosmos qa and Ropes had shown in the **Response to Weaknesses 1**.
> >
> > **Table: Experimental results on Qwen-2-0.5B with More Dataset: Winogrande, Cosmos qa and Ropes**
> > | Method Qwen-2-0.5B   | gsm8k&cosmos_qa&ropes | gsm8k&cosmos_qa&winogrande |
> > |-----|-----|-----|
> > | Fine-tuned Model     | 47.42         | 51.07      |
> > | Weight Avg           | 38.17       | 39.62        |
> > | DARE                 | 38.76      | 42.57  |
> > | Task Arithmetic      | 38.31    | 40.99     |
> > | Ours Layer Level     | **40.12±0.01**   | 43.67±0.02    |
> > | Ours Attn/MLP Level  | 40.08±0.03      | **43.72±0.11**     |
> >
> > | Method Qwen-2-0.5B         | gsm8k&winogrande&ropes | winogrande&cosmos_qa&ropes |
> > |------|--------|--------|
> > | Fine-tuned Model           | 44.62      | 55.75        |
> > | Weight Avg                 | 36.96          | 47.00       |
> > | DARE                       | **45.07**      | 46.97       |
> > | Task Arithmetic            | 38.76     | 47.36         |
> > | Ours Layer Level           | 41.52±0.03    | **48.36±0.08**       |
> > | Ours Attn/MLP Level        | 41.46±0.17      | 48.25±0.05     |
> >
> > For more detailed results, please refer to Table 6 in the paper.
> >
> > **Table: Experimental results on Qwen-2.5-7B**
> > | Merging Two Models      | Math   | Coding | Avg   | &&& | Math   | Translate | Avg   | &&& | Coding | Translate | Avg   |
> > |-------|--------|--------|-------|-----|--------|-----------|-------|-----|--------|-----------|-------|
> > | Fine-tuned Model        | 0.7149 | 0.6036 | 0.6592| &&& | 0.7149 | 0.8583    | 0.7866| &&& | 0.6036 | 0.8583    | 0.7309|
> > | Task Arithmetic         | 0.7513 | 0.6768 | **0.7140**| &&& | 0.7869 | 0.8643    | 0.8256| &&& | 0.6402 | 0.8630    | 0.7516|
> > | Ours Layer Level        | 0.7801 | 0.6402 | 0.7101| &&& | 0.7793 | 0.8655    | 0.8224| &&& | 0.6463 | 0.8659    | **0.7561**|
> > | Ours Attn/MLP Level     | 0.7786 | 0.6463 | 0.7124| &&& | 0.7892 | 0.8658    | **0.8275**| &&& | 0.6402 | 0.8662    | 0.7532|
> >
> > | Merging Three Models    | Math   | Coding | Translate | Avg   |
> > |------|--------|---|----|-----|
> > | Fine-tuned Model        | 0.7149 | 0.6036 | 0.8583    | 0.7256|
> > | Task Arithmetic         | 0.7892 | 0.6524 | 0.8656    | 0.7690|
> > | Ours Layer Level        | 0.8104 | 0.6402 | 0.8665    | 0.7723|
> > | Ours Attn/MLP Level     | 0.8066 | 0.6524 | 0.8664    | **0.7751**|
> >
> >
> > ---
> > **Weaknesses 3:**
> > > The baselines are also somewhat limited, some possible works to consider are e.g. [1], [2] and [3].
> >
> > **Response to Weaknesses 3:**
> > We also appreciate your concern about the baseline. Following your valuable comments, we have conducted more experiments on the new baselines for comparison. The results are presented below. Our methods have shown consistent better performance.
> >
> > **Table: Experimental results with more baseline**
> > | Method       | Math  | Coding | Avg    | &&& | Math  | Coding | Translate | Avg   |
> > |------|-------|--------|------|-----|-------|--------|-----|------|
> > | LLaMA-2-7B    |       |        |     |     |       |        |           |          |
> > | Breadcrumbs[2]     | 18.11 | 18.90  | 18.51     | &&& | 7.96  | 18.90  | 82.78     | 36.55          |
> > | TIES[1]       | 34.49 | 18.90  | 26.69    | &&& | 29.03 | 15.85  | 81.49     | 42.12          |
> > | Consensus TA[3]    | 27.21 | 17.68  | 22.44     | &&& | 26.38 | 17.68  | 82.36     | 42.14          |
> > | Ours Attn/MLP Level    | 35.96 | 18.70  | **27.33**  | &&& | 32.02 | 14.84  | 81.79     | **42.88**      |
> >
> > For each newly added baseline, we adopt the hyperparameters recommended by the original paper and conduct a simple grid search around these hyperparameters.
> >
> >
> >
> > ---

---

> ### Author Response · Authors · 2024-11-19
> **Response by Authors**
>
> ``Part 3/4``
>
> ---
>
>
> **Weaknesses 4:**
> > The main finding of the paper, that is the fact that modules show more linearity than the whole model, obtained as a composition of all those intermediate modules, is not surprising. In fact, if the intermediate modules have non-zero non-linearity scores, the overall non-linearity score will grow as these ones accumulate.
>
> **Response to Weaknesses 4:**
>
> Thank you very much for your valuable feedback and for pointing out important considerations regarding our observations on linearity.
>
> Although our comparative results between the full model and its layers align with the general intuition regarding nonlinearity—that the combination of multiple submodules leads to a cumulative effect of their nonlinearities, resulting in a higher degree of nonlinearity for the combined module—we also observed phenomena that contradict this intuition. Specifically, the MLP, as a submodule within a layer, exhibits higher nonlinearity than the layer itself, and the heads display higher nonlinearity than the attention mechanism, as shown in the table below:
>
> | Level | Non-linearity Score | The Variation |
> |-------|---------------------|-------------------------------------------------------------------------------------|
> | Model | 2.7637 ± 0.3442 | -                                                                                   |
> | Layer | 0.1645 ± 0.4329 | $\downarrow$     (In comparison to Model Level)                                                             |
> | MLP   | 0.2030 ± 0.5655 | $\uparrow$          (In comparison to Layer Level)                                                                |
> | Attn  | 0.0229 ± 0.0107 | $\downarrow$      (In comparison to Layer Level)                                                                |
> | Head  | 0.0727 ± 0.0751 | $\uparrow$      (In comparison to Attn Level)                                                                 |
>
> The origin of these phenomena lies in the fact that the linearity properties we studied in this paper differ from the Traditional Linear Properties, as illustrated in the following table:
>
> |  | Traditional Linear Properties | Linearity Properties Studied in This Paper |
> |-------|-------|-------|
> | Definition | Linear relationship between the difference in output features of different input $f_\theta(x)-f_\theta(x_0)$ and the difference in input $x-x_0$. $f_{\theta}()$denotes model(module) with paramter $\theta$ | Linear relationship between the difference in output features of the model (module) before and after fine-tuning $f_{\theta}(x) - f_{\theta_0}(x)$ and the difference in parameters before and after fine-tuning $\theta - \theta_0$. |
> | Formula Form | $f_\theta(x)-f_\theta(x_0) \approx C(x-x_0)$ where $C$ is a constant determined by $\theta$. | $f_{\theta}(x) - f_{\theta_0}(x) \approx C(\theta - \theta_0)$ where $C$ is a constant determined by $\theta_0$ and $x$. |
> | Determinants | Determined by the model (module) architecture of $\theta$. | Determined by both the alteration introduced by fine-tuning process and the model (module) architecture of $\theta_0$. |
>
>
>
>
> We also present an example to futher illustrate that there is no particularly intuitive connection between the linearity of the module and the linearity of its submodules:
>
> Consider two contiguous submodules, $\theta_1$ and $\theta_2$. Let $\\{\theta_1;\theta_2\\}$ denote the composed module.
>
> If  $\theta_1$ and $\theta_2$ both satisfy the Traditional Linear Properties, it is obvious that the module $\{\theta_1;\theta_2\}$ which formed by these submodules will also adhere to the Traditional Linear Properties.
>
> However, if  $\theta_1$ and $\theta_2$ both satisfy the linearity property studied in this paper, for an arbitrary x, the final output results after applying perturbations $\Delta \theta_1$ and $\Delta \theta_2$ to the two modules is:
>
> |||
> |-------|-------|
> | Input|$x $|
> |Output of the first submodule $\theta_1 + \Delta \theta_1$:|$f_{\theta_1}(x)+C_{\theta_1,x}\Delta\theta_1$|
> |Output of the second submodule $\theta_2 + \Delta \theta_2$:|$f_{\theta_2}(f_{\theta_1}(x)+C_{\theta_1,x}\Delta\theta_1) + C_{\theta_2,f_{\theta_1}(x)+C_{\theta_1,x}\Delta\theta_1}\Delta \theta_2$ ....(1) |
>
> $C_{\theta,x}$ denote a constant determined by $\theta$ and $x.
>
> Now, if $\\{\theta_1;\theta_2\\}$ as a whole satisfies linearity property studied in this paper, then its output should be $f_{\theta_2}(f_{\theta_1}(x)) + C_{\\{\theta_1;\theta_2\\},x} \\{\Delta\theta_1;\Delta\theta_2\\}$ ....(2)  when input $x$.
>
> We can observe a significant difference between formulas (1) and (2) , indicating that there is no particularly intuitive connection between the linearity of the module and the linearity of its submodules,  which also necessitates further experimental and observational analysis, as what we do in this paper.
>
>
>
> ---

---

> > ### Author Response · Authors · 2024-11-19
> > **Response by Authors**
> >
> > ``Part 4/4``
> >
> > ---
> >
> >
> >
> >
> > **Weaknesses 5 and Questions 2:**
> > > Experiments have shown that merging three models exhibits a lower projection distance than merging just two, highlighting the potential of merging even more models. However, the experiments only merge up to three models. Can the trend be confirmed? More experiments would help clarify this aspect.
> >
> > > Maybe add a few more compatible tasks so that merging more than three models reveals some interesting trends as the number of tasks increases. Maybe your framework works even better compared to the baselines when the number of tasks is raised? Or perhaps the advantage becomes less pronounced?
> >
> >
> > **Response to Weaknesses 5 and Questions 2:**
> >
> > Thank you for your valuable feedback and suggestion! Follow your suggestion, we have conducted some additional experiments to further demonstrate the efficiency of our method in scenarios where more models are merged simultaneously.
> >
> > We start with the fail case of code & translate in LLaMA-2-7B mentioned by the reviewer. We increase the number of task-specific models merged simultaneously gradually and calculate the average performance of the merged models.
> >
> > Table :
> > | Avg. Acc                  | Coding & Translate | Coding & Translate & Cosmos_qa | Coding & Translate & Cosmos_qa & Ropes | Coding & Translate & Cosmos_qa & Ropes & Winogrande |
> > |-----|--|-----|-----|--------|
> > | Task Arithmetic   | 0.5111   | 0.5643   | 0.5257    | 0.4666    |
> > | Ours     | 0.4980   | 0.5581    | 0.5382    | 0.5489       |
> > | Gap $\Delta $      | -2.6\%    | -1.1\%     | +2.3\%       | +14.9\%      |
> >
> > In the table, we can observe that as the number of models increases, our method gradually surpasses Task Arithmetic and the performance gap becomes larger, which demonstrates the efficiency of our method particularly when more models are merged simultaneously.
> >
> > ---
> > **Questions 1:**
> > > The requirement of a small set of data per task is reasonable in practice, but what if no data is available? Can you still do something?
> >
> > **Response to Questions 1:**
> > Thank you very much for your feedback. Regarding the issue of data amount, we first conducted an ablation study about the amount of data used in our method, as shown below.
> >
> > **Table : Ablation on the Num of Data**
> > |      | Data Num per Task | 1             | 3             | 10            | 30            | 50            |
> > |-----|-----|----|---|-----|---|-----|
> > |    |   **Layer Level Merge Three Models**   |    | |    |    |   |
> > |     | Math      | 29.92 ± 1.53  | 31.19 ± 0.36  | 30.83 ± 0.28  | 30.50 ± 0.54  | 30.81 ± 0.48  |
> > |   | Coding    | 16.87 ± 1.75  | 15.24 ± 0.50  | 15.65 ± 0.57  | 15.85 ± 0.50  | 15.65 ± 0.29  |
> > |   | Translate   | 81.78 ± 0.09  | 81.67 ± 0.02  | 81.72 ± 0.02  | 81.68 ± 0.05  | 81.80 ± 0.03  |
> > |   |  **Attn/MLP Level Merge Three Models**    |     |      |     |     |  |
> > | | Math  | 32.17 ± 0.73  | 32.78 ± 0.68  | 32.85 ± 0.60  | 32.02 ± 0.31  | 32.68 ± 0.06  |
> > |    | Coding  | 15.24 ± 0.86  | 14.63 ± 0.01  | 14.63 ± 0.50  | 14.84 ± 0.57  | 14.43 ± 0.29  |
> > |   | Translate    | 81.84 ± 0.05  | 81.74 ± 0.04  | 81.72 ± 0.06  | 81.79 ± 0.03  | 81.79 ± 0.05  |
> >
> > From the table, we can see that our method shows consistent performance across different data quantities, sometimes achieving good results with as few as 3 data per task. As a result, if one wishes to use our method in a situation of absolutely no data, we believe that simply synthesizing a few data for the corresponding task through manual methods or with the help of LLMs is feasible.
> >
> > ---
> >
> > ---
> > **Questions 3:**
> > > Minor suggestion: In Property 2, I would really avoid using footnotes over equations as these look like powers instead.
> >
> > **Response to Questions 3:**
> > Thank you very much for your suggestions! We have moved this footnote to the description of the formula in our revision.
> >
> > ---
> >
> >
> > **References**
> >
> >
> > [1] Yadav, Prateek, et al. "Ties-merging: Resolving interference when merging models." Advances in Neural Information Processing Systems 36 (2024).
> >
> > [2] Davari, MohammadReza, and Eugene Belilovsky. "Model breadcrumbs: Scaling multi-task model merging with sparse masks." ECCV 2024.
> >
> > [3] Wang, Ke, et al. "Localizing Task Information for Improved Model Merging and Compression." Forty-first International Conference on Machine Learning.
> >
> >
> > ---
> >
> > Please let us know if our response has addressed your concerns or if you have any further questions. We would be grateful for the opportunity to continue discussing with you. Thank you once again for your valuable feedback!

---

> > > ### Comment · Reviewer_g3he · 2024-11-23
> > >
> > > I thank the authors for their rebuttal.
> > > All my concerns have been addressed except the gains which still seem marginal.
> > > I am raising my score to 6.

---

> > > > ### Author Response · Authors · 2024-11-23
> > > >
> > > > Dear Reviewer g3he,
> > > >
> > > > We are delighted to hear that your concerns have been addressed. We sincerely appreciate your decision to raise the score.
> > > >
> > > > Thank you once again for taking the time to review our paper!
> > > >
> > > > Best regards,
> > > >
> > > > The Authors of Paper #3017

---

### Official Review · Reviewer_F2K5 · 2024-11-03

**Soundness:** 3
**Presentation:** 3
**Contribution:** 2
**Rating:** 6
**Confidence:** 4

**Summary:**

This paper conducted a statistical analysis to examine the linearity of model-level and decomposed submodule-level components. The authors discovered that submodules (e.g., layers, self-attentions, and MLPs) demonstrate significantly higher linearity compared to the overall model. Based on this finding, they introduced a new model merging strategy that independently merges these submodules. Experimental results indicate that the proposed method surpasses the standard task arithmetic approach and other baselines across various model scales and tasks.

**Strengths:**

1. The paper is elegantly written and easy to comprehend. The motivation, proposed method, and experiments are all well elucidated.
2. The authors focus on an important issue in model merging, and proposed a principled approach to address this issue.
3. The authors offer detailed explanations of the proposed method and experimental outcomes in the appendix, which is highly commendable.

**Weaknesses:**

1. The observation that submodules exhibit higher linearity than the overall model may be deemed rather obvious. Since the overall model is constructed from these submodules, the nonlinearity of each submodule contributes to the overall model's nonlinearity. Consequently, the comparison results presented in Figure 1 and Figure 2 are unsurprising. The linearity of the model is fundamentally inferior to that of its submodules.

2. Building upon the first comment, the proposed method could be interpreted as a straightforward extension of previous global linearization approaches. The authors introduce a layerwise linearization version, which has been explored in prior studies, such as Git Re-Basin [1], Zipit! [2], and MuDSC [3]. It would be beneficial if the authors could provide comparisons with these works.

3. Furthermore, it is conceivable to decompose the model into more finely-grained components, such as different groups, channels, or even neurons. I posit that the linearity of these components is more pronounced than that of layer-level submodules. However, this would necessitate a greater number of merging parameters, similar to the method proposed in this paper. Could the authors engage in discussions regarding this aspect?

4. The three properties outlined in Section 2 are somewhat redundant and could benefit from simplification.

5. The performance improvement is not so impressive. As the proposed method adopts a layerwise merging strategy, it is expected to be much more performant than others. However, experiments demonstrate that the proposed method is still outperformed by prior methods. For example, coding & translate, math & coding in Table 1 and Table 2.


### References
[1] Git Re-Basin: Merging Models modulo Permutation Symmetries, ICLR 2023.

[2] Zipit!: Merging Models from Different Tasks without Training. NeurIPS 2023.

[3] Training-Free Pretrained Model Merging, CVPR 2024.

**Questions:**

Please see the Weaknesses

---

> ### Author Response · Authors · 2024-11-19
> **Response by Authors**
>
> ``Part 1/4``
>
> ---
>
> **Response by Authors:**
>
> We would like to express our sincere gratitude for your valuable comments and feedback. It is our pleasure to address your concerns and accept your insightful suggestions. We would like to acknowledge the time and effort you have invested in reviewing our work. Below, we provide a detailed response to each comment.
>
> ---
> **Weaknesses 1:**
> > The observation that submodules exhibit higher linearity than the overall model may be deemed rather obvious. Since the overall model is constructed from these submodules, the nonlinearity of each submodule contributes to the overall model's nonlinearity. Consequently, the comparison results presented in Figure 1 and Figure 2 are unsurprising. The linearity of the model is fundamentally inferior to that of its submodules.
>
> **Response to Weaknesses 1:**
> Thank you very much for your valuable feedback and for pointing out important considerations regarding our observations on linearity.
>
> Although our comparative results between the full model and its layers align with the general intuition regarding nonlinearity—that the combination of multiple submodules leads to a cumulative effect of their nonlinearities, resulting in a higher degree of nonlinearity for the combined module—we also observed phenomena that contradict this intuition. Specifically, the MLP, as a submodule within a layer, exhibits higher nonlinearity than the layer itself, and the heads display higher nonlinearity than the attention mechanism, as shown in the table below:
>
>
> | Level | Non-linearity Score | The Variation |
> |--|--|--|
> | Model | 2.7637 ± 0.3442 | -    |
> | Layer | 0.1645 ± 0.4329 | $\downarrow$ (In comparison to Model Level)  |
> | MLP   | 0.2030 ± 0.5655 | $\uparrow$ (In comparison to Layer Level)  |
> | Attn  | 0.0229 ± 0.0107 | $\downarrow$ (In comparison to Layer Level)  |
> | Head  | 0.0727 ± 0.0751 | $\uparrow$ (In comparison to Attn Level)  |
>
> The origin of these phenomena lies in the fact that the linearity properties we studied in this paper differ from the Traditional Linear Properties, as illustrated in the following table:
>
>
> |  | Traditional Linear Properties | Linearity Properties Studied in This Paper |
> |-------|-------|-------|
> | Definition | Linear relationship between the difference in output features of different input $f_\theta(x)-f_\theta(x_0)$ and the difference in input $x-x_0$. $f_{\theta}()$denotes model(module) with paramter $\theta$ | Linear relationship between the difference in output features of the model (module) before and after fine-tuning $f_{\theta}(x) - f_{\theta_0}(x)$ and the difference in parameters before and after fine-tuning $\theta - \theta_0$. |
> | Formula Form | $f_\theta(x)-f_\theta(x_0) \approx C(x-x_0)$ where $C$ is a constant determined by $\theta$. | $f_{\theta}(x) - f_{\theta_0}(x) \approx C(\theta - \theta_0)$ where $C$ is a constant determined by $\theta_0$ and $x$. |
> | Determinants | Determined by the model (module) architecture of $\theta$. | Determined by both the alteration introduced by fine-tuning process and the model (module) architecture of $\theta_0$. |
>
>
> We also present an example to futher illustrate that there is no particularly intuitive connection between the linearity of the module and the linearity of its submodules:
>
> Consider two contiguous submodules, $\theta_1$ and $\theta_2$. Let $\\{\theta_1;\theta_2\\}$ denote the composed module.
>
> If  $\theta_1$ and $\theta_2$ both satisfy the Traditional Linear Properties, it is obvious that the module $\{\theta_1;\theta_2\}$ which formed by these submodules will also adhere to the Traditional Linear Properties.
>
> However, if  $\theta_1$ and $\theta_2$ both satisfy the linearity property studied in this paper, for an arbitrary x, the final output results after applying perturbations $\Delta \theta_1$ and $\Delta \theta_2$ to the two modules is:
>
> |||
> |-------|-------|
> | Input|$x $|
> |Output of the first submodule $\theta_1 + \Delta \theta_1$:|$f_{\theta_1}(x)+C_{\theta_1,x}\Delta\theta_1$|
> |Output of the second submodule $\theta_2 + \Delta \theta_2$:|$f_{\theta_2}(f_{\theta_1}(x)+C_{\theta_1,x}\Delta\theta_1) + C_{\theta_2,f_{\theta_1}(x)+C_{\theta_1,x}\Delta\theta_1}\Delta \theta_2$ ....(1) |
>
> $C_{\theta,x}$ denote a constant determined by $\theta$ and $x.
>
> Now, if $\\{\theta_1;\theta_2\\}$ as a whole satisfies linearity property studied in this paper, then its output should be $f_{\theta_2}(f_{\theta_1}(x)) + C_{\\{\theta_1;\theta_2\\},x} \\{\Delta\theta_1;\Delta\theta_2\\} $ ....(2)  when input $x$.
>
> We can observe a significant difference between formulas (1) and (2) , indicating that there is no particularly intuitive connection between the linearity of the module and the linearity of its submodules,  which also necessitates further experimental and observational analysis, as what we do in this paper.
>
>
>
> ---

---

> > ### Author Response · Authors · 2024-11-19
> > **Response by Authors**
> >
> > ``Part 2/4``
> >
> > ---
> >
> >
> > **Weaknesses 2:**
> > > Building upon the first comment, the proposed method could be interpreted as a straightforward extension of previous global linearization approaches. The authors introduce a layerwise linearization version, which has been explored in prior studies, such as Git Re-Basin [1], Zipit! [2], and MuDSC [3]. It would be beneficial if the authors could provide comparisons with these works.
> >
> > **Response to Weaknesses 2:**
> > Thank you for bringing these works [1,2,3] to our attention. We have found that [1,2,3] are indeed highly worthy of research in the field of computer vision and have some relevance to our approach. As mentioned in the comments, we will add the discussion in the related work section. We present the comaparison between these works and ours in the following table.
> >
> > |Comparison Table|Applicable Module|The Type of Linearity Relied upon during the Merging|Key method for merging|
> > |--|--|--|--|
> > |Git Re Basin [1]|Linear Layer (full connected layer) in Network |Traditional Linearity Properties |Combinatorial optimization |
> > |Zipit! [2] |Linear Layer (full connected layer) in Network |Traditional Linearity Properties |Align features by similarity in the activation space|
> > |MuDSC [3]|Linear Layer (full connected layer) in Network |Traditional Linearity Properties |Align units by similarity in both the parameter space and the activation space|
> > |Ours|Each submodule level (including layer,attn and MLP) that exibit Linearity introduced in this paper|Linearity Properties Studied in This Paper |Closed-form solution of optimal merging weights|
> >
> > As shown in the table, all these methods relies on the Traditional Linearity Property, and can only apply to linear layers. In summary, we believe that our method has several novel contributions that make it distinct from Git Re Basin[1], Zipit![2] and MuDSC[3].
> >
> >
> > ---

---

> > > ### Author Response · Authors · 2024-11-19
> > > **Response by Authors**
> > >
> > > ``Part 3/4``
> > >
> > > ---
> > >
> > > **Weaknesses 3:**
> > > > Furthermore, it is conceivable to decompose the model into more finely-grained components, such as different groups, channels, or even neurons. I posit that the linearity of these components is more pronounced than that of layer-level submodules. However, this would necessitate a greater number of merging parameters, similar to the method proposed in this paper. Could the authors engage in discussions regarding this aspect?
> > >
> > > **Response to Weaknesses 3:**
> > > Thank you very much for your valuable suggestions. We will add the following discussion in our revision.
> > >
> > > Further decomposing the model into more finely-grained components is indeed an intuitive and promising direction for exploration. In the paper, we conducted some related experiments. Based on the structure of the LLM model, we further divided the attention layer into individual heads, independently estimating their linearity and ultimately testing their performance, but revealing unstable outcomes at this level (as shown in table the below).
> > >
> > > **Table: Ablation results of merging fine-tuned models by different levels in our method**
> > > | **LLaMA-2-7B**   | **Math & Coding**   | **Math & Translate**  | **Coding & Translate**  | **Math & Coding & Translate**  |
> > > |---|----|---|-----|--|
> > > | Ours Layer Level      | 27.18±0.04 | 49.79±0.43  | 50.64±0.64  | 42.68±0.26 |
> > > | Ours Attn/MLP Level   | 27.33±0.10  | 55.86±0.48   | 49.80±0.23   | 42.88±0.27  |
> > > | Ours Head/MLP Level   | 13.08±9.33  | 14.17±1.14   | 41.14±12.39  | 8.16±1.30  |
> > >
> > > | **Llama-2-13B**   | **Math & Coding**  | **Math & Translate** | **Coding & Translate**   | **Math & Coding & Translate**  |
> > > |---|--|----|--|----|
> > > | Ours Layer Level      | 34.47±0.17   | 66.08±0.14  | 52.42±0.15  | 50.80±0.17  |
> > > | Ours Attn/MLP Level   | 35.11±0.29   | 66.11±0.11   | 52.51±0.26   | 51.05±0.16  |
> > > | Ours Head/MLP Level   | 12.91±15.60   | 30.17±19.36    | 13.71±3.50  | 22.59±19.65  |
> > >
> > > Here we propose a potential explanation for the occurrence of this situation. Numerous studies [4,5,6] suggest that each head in the model performs different roles during inference. Following fine-tuning across various tasks, certain heads may adopt distinct functions. However, our objective function may necessitate a single head to simultaneously average among multiple heads with diverse functionalities. This requirement can be quite challenging for an individual head and may ultimately result in a collapse of functionality in the merged head.
> > >
> > > However, this may not imply that the linearity of the head or more finely-grained components is unhelpful for the merging process. As we mentioned in our response to weakness 1, the relationship between the linearity of the module and the linearity of its submodules is not straightforward; therefore, further experiments and analyses are required. Following the reviewer's comments, we will further explore whether there are better ways to address the aforementioned issues in future work. For instance, based on the architecture of the LLM, we could further decompose each module into individual linear layers (decompose attention into qkvo projection linear layers, and MLP into two linear layers). Alternatively, we could incorporate interpretability methods [4, 5] to improve understanding and identify which head is most relevant to a specific task.
> > >
> > > ---
> > > **Weaknesses 4:**
> > > > The three properties outlined in Section 2 are somewhat redundant and could benefit from simplification.
> > >
> > > **Response to Weaknesses 4:**
> > > We greatly appreciate your valuable suggestions. There is indeed a certain degree of repetition in the three Properties concepts. In our revision, we will merge Properties 1 (for model) and Properties 2 (for module) into a more general Properties.
> > > ($f(x;\theta_{0}+\alpha \tau ) \approx f(x;\theta_{0})+\alpha \Delta f(x;\theta_{0}+\tau )$  Now, $f(;\theta)$ can represent a model or module.)
> > >
> > >
> > >
> > > ---

---

> > > > ### Author Response · Authors · 2024-11-19
> > > > **Response by Authors**
> > > >
> > > > ``Part 4/4``
> > > >
> > > > ---
> > > >
> > > > **Weaknesses 5:**
> > > > > The performance improvement is not so impressive. As the proposed method adopts a layerwise merging strategy, it is expected to be much more performant than others. However, experiments demonstrate that the proposed method is still outperformed by prior methods. For example, coding & translate, math & coding in Table 1 and Table 2.
> > > >
> > > > **Response to Weaknesses 5:**
> > > > Thank you for your valuable feedback! We have conducted some additional experiments to further demonstrate the efficiency of our method.
> > > >
> > > > Firstly, to demonstrate the advantages of our method in scenarios where more models are merged simultaneously, we start with the fail case of code & translate in LLaMA-2-7B mentioned by the reviewer. We increase the number of task-specific models merged simultaneously gradually and calculate the average performance of the merged models.
> > > >
> > > > **Table : Ablation study on Number of Model to be Merged**
> > > > | Avg. Acc                  | Coding & Translate | Coding & Translate & Cosmos_qa | Coding & Translate & Cosmos_qa & Ropes | Coding & Translate & Cosmos_qa & Ropes & Winogrande |
> > > > |--|--|--|--|--|
> > > > | Task Arithmetic  | 0.5111 | 0.5643  | 0.5257 | 0.4666 |
> > > > | Ours  | 0.4980 | 0.5581 | 0.5382 | 0.5489  |
> > > > | Gap $\Delta $  | -2.6\%  | -1.1\%  | +2.3\%| +14.9\%|
> > > >
> > > > In the table, we can observe that as the number of models increases, our method gradually surpasses Task Arithmetic and the performance gap becomes larger, which demonstrates the efficiency of our method particularly when more models are merged simultaneously.
> > > >
> > > >
> > > > We also have conducted additional experiments utilizing the new datasets. The results are presented below, demonstrating that our methods consistently exhibit impressive performance.
> > > >
> > > > **Table: Experimental results on LLaMA-2-7B with More Dataset: Winogrande, Cosmos qa and Ropes**
> > > > | Merging Two Models  | Winogrande | Cosmos qa | Avg   | &&& | Winogrande | Ropes  | Avg   | &&& | Cosmos qa | Ropes  | Avg   |
> > > > |---|---|--|--|---|--|--|---|---|--|----|---|
> > > > | Fine-tuned Model | 0.8226     | 0.8472    | 0.8349| &&& | 0.8226     | 0.7542 | 0.7884| &&& | 0.8472    | 0.7542 | 0.8007|
> > > > | Task Arithmetic | 0.7222     | 0.7185    | 0.7203| &&& | 0.7214     | 0.5428 | 0.6321| &&& | 0.7348    | 0.5875 | 0.6611|
> > > > | Ours Attn/MLP Level   | 0.6842     | 0.8418    | **0.7630**| &&& | 0.6947     | 0.6998 | **0.6972**| &&& | 0.7307    | 0.6859 | **0.7083**|
> > > >
> > > > | Merging Three Models  | Winogrande | Cosmos qa | Ropes  | Avg   |
> > > > |----|-----|---|---|----|
> > > > | Fine-tuned Model  | 0.8226     | 0.8472    | 0.7542 | 0.808 |
> > > > | Task Arithmetic   | 0.7125     | 0.6413    | 0.5585 | 0.6374|
> > > > | Ours Attn/MLP Level | 0.7149     | 0.6870    | 0.5670 | **0.6563**|
> > > >
> > > >
> > > > We also provide potential explanations for the few fail cases mentioned by the reviewer (such as coding & translation in LLaMA-2-7B). Our loss design is intended to ensure that the output features of the merged submodule for data from different tasks closely align with the output features of the corresponding task-specific models. To visualize this, we employ t-SNE to observe the distribution of features output by each module when different data inputs are provided. (Please refer to this anonymous link https://anonymous.4open.science/r/rebuttal-tsne-result-select for the sample of t-SNE result, as the OpenReview website does not support direct image uploads.)
> > > >
> > > > We found that the output feature distributions for code data and translate data on various modules in LLaMA-2-7B tend to be relatively close overall (see table below), and sometimes even intertwine, which contradicts the goal of our loss design and may have led to this fail case. These fail cases occur in only a few situations, with over 90% of the results demonstrating the efficiency of our method.
> > > >
> > > > **Table : Avg. Norm Distance of Output Feature in all t-SNE results**
> > > > |    | Avg. Norm Distance of Output Feature |
> > > > |----|--|
> > > > | Math vs. Coding         | 0.3475 |
> > > > | Math vs. Translate      | 0.3758 |
> > > > | Coding vs. Translate    | 0.2765   |
> > > >
> > > > In our future work, we will focus on improving these rare fail cases. We plan to explore more efficient methods that utilize linearity, such as incorporating second-order information as suggested by reviewer SeC4, in order to further enhance the overall performance of our approach.
> > > >
> > > > ---
> > > >
> > > >
> > > > **References**
> > > >
> > > > [1] Git Re-Basin: Merging Models modulo Permutation Symmetries, ICLR 2023.
> > > >
> > > > [2] Zipit!: Merging Models from Different Tasks without Training. NeurIPS 2023.
> > > >
> > > > [3] Training-Free Pretrained Model Merging, CVPR 2024.
> > > >
> > > > [4] Meng, Kevin, et al. Locating and editing factual associations in GPT.  NIPS2022
> > > >
> > > > [5] Zhang, Fred, and Neel Nanda. Towards best practices of activation patching in language models: Metrics and methods. ICLR2024
> > > >
> > > >
> > > > ---
> > > >
> > > >
> > > > We hope that our response has addressed your concerns. If you have any further questions or suggestions, please do not hesitate to let us know. Thank you once again for your valuable feedback!

---

> > > > > ### Comment · Reviewer_F2K5 · 2024-11-25
> > > > >
> > > > > Thank you for your thorough and thoughtful response.
> > > > >
> > > > > Nevertheless, I feel that my principal concerns remain unresolved. In the reply to Weakness #1, the authors present a table to support the claim *the MLP, as a submodule within a layer, exhibits greater nonlinearity than the layer itself, and that the heads demonstrate higher nonlinearity than the attention mechanism*. However, the results clearly reveal that the standard deviation of the nonlinearity score for the MLP is significantly larger than that for the layer. Similarly, the standard deviation of the nonlinearity score for the heads surpasses that of the attention mechanism. Given these results, it is difficult to maintain that the findings contradict conventional intuition. Furthermore, the subsequent analysis comparing Traditional Linear Properties vs the Linearity Properties explored in this paper fails to delineate any substantial differences.
> > > > >
> > > > > Taking all factors into account, I am sorry that I remain my initial rating of the paper.

---

> > > > > > ### Author Response · Authors · 2024-11-25
> > > > > >
> > > > > > We greatly appreciate receiving your feedback! We are delighted to hear that your other concerns have been addressed.
> > > > > >
> > > > > > To further address your concern regarding the standard deviation of the Non-linearity Score, we have provided below **the raw Non-linearity Scores** of the Llama-2-7B model fine-tuned on GSM8K as example. (Due to the large number of Heads, we presents the results of the first two Heads from each Attn here.)
> > > > > >
> > > > > > | Raw Non-Linearity Score | Layer 0 | 1 | 2 | 3 | 4 | 5 | 6 | 7 | 8 | 9 | 10 | 11 | 12 | 13 | 14 | 15 | 16 | 17 | 18 | 19 | 20 | 21 | 22 | 23 | 24 | 25 | 26 | 27 | 28 | 29 | 30 | 31 |
> > > > > > | --|--| --| --| --| --| --| --| --| --| --| --| --| --| --| --| --| --| --| --| --| --| --| --| --| --| --| --| --| --| --| --| --|
> > > > > >  | Model Level | 3.1798 |
> > > > > >  |Layer Level | 0.0848 | 0.032 | 0.0142 | 0.1029 | 0.0164 | 0.0635 | 0.2371 | 0.0209 | 0.1009 | 0.1157 | 0.5707 | 0.0196 | 0.0157 | 0.0198 | 0.0261 | 0.0638 | 0.1389 | 0.0472 | 0.0936 | 0.0185 | 0.0197 | 0.019 | 0.021 | 0.1268 | 0.0159 | 0.0233 | 0.0299 | 0.1748 | 0.0353 | 0.0345 | 0.041 | 0.0885 |
> > > > > >  | MLP Level | 0.0962 | 0.0388 | 0.0134 | 0.0205 | 0.062 | 0.0167 | 0.0303 | 0.063 | 0.0182 | 0.0167 | 0.0451 | 0.0202 | 0.0311 | 0.0901 | 0.0384 | 0.0601 | 0.018 | 0.0242 | 0.0596 | 0.0221 | 1.0576 | 0.0497 | 0.0212 | 0.0704 | 0.0182 | 0.0783 | 0.0195 | 0.018 | 1.9705 | 0.0184 | 0.041 | 0.0917 | 0.0962 | 0.0388 | 0.0134 | 0.0205 | 0.062 | 0.0167 | 0.0303 | 0.063 | 0.0182 | 0.0167 | 0.0451 | 0.0202 | 0.0311 | 0.0901 | 0.0384 | 0.0601 | 0.018 | 0.0242 | 0.0596 | 0.0221 | 1.0576 | 0.0497 | 0.0212 | 0.0704 | 0.0182 | 0.0783 | 0.0195 | 0.018 | 1.9705 | 0.0184 | 0.041 | 0.0917 |
> > > > > >  | Attn Level | 0.0214 | 0.0146 | 0.0314 | 0.0217 | 0.0197 | 0.0362 | 0.0215 | 0.0174 | 0.0224 | 0.0289 | 0.0209 | 0.0114 | 0.0162 | 0.0257 | 0.0168 | 0.0163 | 0.015 | 0.0272 | 0.0033 | 0.0164 | 0.0145 | 0.0188 | 0.0169 | 0.0363 | 0.023 | 0.0252 | 0.0233 | 0.0302 | 0.0258 | 0.0138 | 0.0031 | 0.0282 |
> > > > > >  | Head_0 Level | 0.0381 | 0.0911 | 0.0353 | 0.2023 | 0.0334 | 0.1672 | 0.12 | 0.055 | 0.0549 | 0.0768 | 0.0214 | 0.0654 | 0.1203 | 0.0692 | 0.0671 | 0.0382 | 0.0336 | 0.0467 | 0.0263 | 0.0419 | 0.0375 | 0.0776 | 0.1595 | 0.0031 | 0.0281 | 0.0033 | 0.0886 | 0.0323 | 0.0441 | 0.0391 | 0.0404 | 0.0302 |
> > > > > >  | Head_1 Level | 0.0275 | 0.0186 | 0.041 | 0.1449 | 0.0486 | 0.0267 | 0.0605 | 0.0996 | 0.0144 | 0.0774 | 0.1392 | 0.0394 | 0.0731 | 0.0392 | 0.0083 | 0.0349 | 0.0795 | 0.0684 | 0.0495 | 0.07 | 0.0294 | 0.0352 | 0.0456 | 0.0485 | 0.1168 | 0.0381 | 0.0195 | 0.0204 | 0.0673 | 0.1032 | 0.0742 | 0.0215 |
> > > > > >
> > > > > > For your convenience, we have conducted a simple statistical analysis of this table.
> > > > > >
> > > > > > |||
> > > > > > |- |- |
> > > > > > |The proportion of MLP that exceed their corresponding Layer in terms of Non-linearity Score |50% |
> > > > > > |The proportion of Head_0 that exceed their corresponding Attn in terms of Non-linearity Score | 93.75% |
> > > > > > |The proportion of Head_1 that exceed their corresponding Attn in terms of Non-linearity Score |81.25% |
> > > > > >
> > > > > > We can directly observe that, as a submodule of Layer, **50% of MLPs have Non-linearity Scores that exceed their actual corresponding Layer**. At the Head level, **over 80% to 90% of Heads have Non-linearity Scores that exceed their actual corresponding Attn**. These results directly contradict the conventional intuition that "Submodules exhibit higher linearity than the overall Module."
> > > > > >
> > > > > > We also appreciate your meticulous observations. We believe that the larger standard deviation of submodules Non-linearity Score does not leads to limited our statement. On the contrary, it further highlights that the connection between module's linearity and its submodules' linearity is not directly and easily analyzable, which resonates with the results of examples of linearity we provided in our previous response.
> > > > > >
> > > > > > Then, the subsequent analysis comparing Traditional Linear Properties with the Linearity Properties explored in this paper is intended to highlight the significant differences between the two, which indicates that there is no obvious intuitive connection between the linearity of the module and the linearity of its submodules like Traditional Linear Properties.
> > > > > >
> > > > > > We sincerely hope that our response has addressed your remaining concerns. We are looking forward to your feedback and further discussions!

---

> > > > > > ### Author Response · Authors · 2024-11-28
> > > > > >
> > > > > > Dear Reviewer F2K5,
> > > > > >
> > > > > > Thank you for your valuable time and constructive comments! We have made effort to address your remaining concern that **the standard deviation of the Non-linearity Score may affect our statement about the conventional intuition**.  For your convenience, here is a summary of our responses:
> > > > > >
> > > > > > * We have provided **the raw Non-linearity Scores** of the Llama-2-7B model fine-tuned on GSM8K as example and conducted a simple statistical analysis.
> > > > > >
> > > > > > * From the raw data and analysis, **we can observe a considerable number of examples that directly contradict the conventional intuition** that "Submodules exhibit higher linearity than the overall Module."
> > > > > >
> > > > > > * The subsequent analysis, which compares Traditional Linear Properties with the Linearity Properties explored in this paper, aims to **underscore the significant differences between the two**, which may account for the aforementioned phenomena.
> > > > > >
> > > > > > * **We have revised our manuscript based on your valuable comments** before the revision submission system closes.
> > > > > >
> > > > > > We sincerely hope our responses can address your concern. If you have any additional concerns or comments that we may have overlooked, we would greatly appreciate any further feedback from you to help us enhance our work.
> > > > > >
> > > > > > Best regards,
> > > > > >
> > > > > > Authors of #3017

---

> > > > > > > ### Comment · Reviewer_F2K5 · 2024-11-28
> > > > > > >
> > > > > > > Thank you for your efforts in addressing the concerns. I am satisfied with the current version and willing to raise the score accordingly.

---

> > > > > > > > ### Author Response · Authors · 2024-11-28
> > > > > > > >
> > > > > > > > Dear Reviewer F2K5,
> > > > > > > >
> > > > > > > > Very glad to hear that your concerns have been addressed!
> > > > > > > >
> > > > > > > > Thanks for raising the score and taking the time to review our paper!
> > > > > > > >
> > > > > > > > Best regards,
> > > > > > > >
> > > > > > > > Authors of #3017

---

### Official Review · Reviewer_SeC4 · 2024-11-03

**Soundness:** 3
**Presentation:** 3
**Contribution:** 3
**Rating:** 6
**Confidence:** 3

**Summary:**

This paper shows that by identifying submodules that exhibit linearity, more granular and effective merging can be achieved. After defining these submodules, they approximate the true objective with a linear interpolation, which holds when the submodules behave linearly. The objective is defined as minimizing the loss on the joint dataset by adjusting the weighting coefficients $\alpha$. This approach outperforms existing parameter merging methods, such as Task Arithmetic. They also perform insightful ablations to determine the optimal level of merging granularity—whether at the model, MLP/attention, or head level.

**Strengths:**

Clear presentation, and interesting proposition of understanding which set of weights are more amenable to being merged. Novel idea of relaxing the original optimization with a linear approximation, which provides a fast and closed form solution. Good experimentation, and valuable ablation study on granularity of merging. The paper also shows impressive results with only 30 datapoints per task, making it highly efficient in terms of data and computational demands. It performs well across different scales like Llama-2-7B and Llama-2-13B, highlighting scalability for both small and large models. The approach is flexible, enabling merging at different granularities such as layer-level and attention/MLP-level, and delivers consistent performance across these structures.

**Weaknesses:**

The authors overlook referencing Mixture of Experts (MoE) merging approaches, such as in https://arxiv.org/abs/2402.00433, where merging also involves linearly weighting submodules. However, in traditional MoE, this weighting occurs at the activation level, rather than directly on model parameters. Since the authors allow for optimization of the weighting coefficients on the joint dataset, then the idea in this paper can be a useful baseline to run: https://arxiv.org/pdf/2410.13025.

That said, it’s unclear why the proposed method would surpass a router-based approach in performance, as a trained router can achieve task-specific specialization without approximations to the objective. While the linear interpolation method is efficient, a router's task-based dynamic routing could potentially allow finer-grained adjustments in response to specific inputs or task demands, leveraging the submodule structures without relying on assumptions of linearity. Adding comparisons or addressing these differences in the paper would clarify the potential advantages and trade-offs between the approaches.

**Questions:**

An interesting direction for improvement could be incorporating second-order information into the optimization process rather than updating each weighting parameter independently. Currently, the approach adjusts the coefficients linearly, assuming each weighting parameter can be optimized in isolation. However, second-order information, such as Hessian-based adjustments or other curvature-aware methods, could potentially capture interactions between weighting parameters, especially when submodules exhibit interdependencies. This is why routing methods could be more powerful, as by iteratively optimizing each layer they can better account for submodule interdependency.

---

> ### Author Response · Authors · 2024-11-19
> **Response by Authors**
>
> `Part 1/2`
>
> ---
>
>
> **Response by Authors**
>
> Thank you for taking the time to review our work and provide valuable comments! We appreciate your comments and suggestions, which have helped us improve our paper. Please find below our detailed responses.
>
> ---
>
> **Weaknesses:**
> > The authors overlook referencing Mixture of Experts (MoE) merging approaches, such as in https://arxiv.org/abs/2402.00433, where merging also involves linearly weighting submodules. However, in traditional MoE, this weighting occurs at the activation level, rather than directly on model parameters. Since the authors allow for optimization of the weighting coefficients on the joint dataset, then the idea in this paper can be a useful baseline to run: https://arxiv.org/pdf/2410.13025. That said, it’s unclear why the proposed method would surpass a router-based approach in performance, as a trained router can achieve task-specific specialization without approximations to the objective. While the linear interpolation method is efficient, a router's task-based dynamic routing could potentially allow finer-grained adjustments in response to specific inputs or task demands, leveraging the submodule structures without relying on assumptions of linearity. Adding comparisons or addressing these differences in the paper would clarify the potential advantages and trade-offs between the approaches.
>
> **Response to Weaknesses:**
> Thank you for bringing these works [1,2] to our attention. We have found that [1,2] are indeed highly worthy of research and discussion regarding the similarities and differences with our method. As mentioned in the comments, we have added the following discussion in the Related Work section. We present our discussion with a table here.
>
>
> | Categories                                      | Ours                 | BTX [3] (std MoE)                              | WEMoE [1]                                     | CAT [2]                                          |
> |-------------------------------------------------|----------------------|------------------------------------------------|-----------------------------------------------|--------------------------------------------------|
> | Requirements for the Model to be Merged         | Same pretrain model, conventional fine-tuning with different task. | Same pretrain model, conventional fine-tuning with different task.|                 Same pretrain model, conventional fine-tuning with different task.                              |                      Same pretrain model, LoRA fine-tuning with different task.                             |
> | Which Part to be Merged                         | All modules          | All modules except MLP                         | All modules except MLP, The MLP will be merged during inference. | All LoRA's Parameter                              |
> | Hyperparameters or Additional Parameters Required for Merging | Weights for each submodules (such as Layer/Attn/MLP) | Router Network for each MLP                    | Router Network for each MLP                    | Weights for each Lora Martix                      |
> | Optimization Methods                            | Closed-form Solution (Training-free) | Training                                        | Training                                      | Training                                         |
> | The Architecture of the Merged Model            | Same as Model to be Merged | MoE Architecture (Router for mlp's output Merging Weights) | MoE Architecture (Router for Parameter Merging Weights) | Same as Model to be Merged                      |
>
>
>
> In our approach, merging is performed directly, which may result in different merging outcomes due to the selection of varying data. Our method requires only minimal data and does not necessitate training, yielding a model with a standard architecture. In comparison to our method, WEMoE constructs a MoE architecture model and employs a Router during inference to determine the merging weights for each MLP based on the current input, which also utilizes a linear merging form. As a result, WEMoE may get better merging weights and achieve better results but incurs additional training and inference costs.
>
>
> In summary, we believe that our methods has several novel contributions that make it distinct from WEMoE. Meanwhile, we acknowledge the importance of [1,2] and its relevance to our work. Accordingly, we have added the discussion in our revision to highlight the contribution and difference between our methods and [1,2].
>
>
> ---

---

> > ### Author Response · Authors · 2024-11-19
> > **Response by Authors**
> >
> > `Part 2/2`
> >
> > ---
> >
> > **Questions:**
> > > An interesting direction for improvement could be incorporating second-order information into the optimization process rather than updating each weighting parameter independently. Currently, the approach adjusts the coefficients linearly, assuming each weighting parameter can be optimized in isolation. However, second-order information, such as Hessian-based adjustments or other curvature-aware methods, could potentially capture interactions between weighting parameters, especially when submodules exhibit interdependencies. This is why routing methods could be more powerful, as by iteratively optimizing each layer they can better account for submodule interdependency.
> >
> >
> > **Response to Questions:**
> > Thank you for your valuable suggestion. Incorporating second-order information is a very valuable suggestion. We have added the following discussion in our revision.
> >
> > In our initial setup, considering that the original form of Task Arithmetic is first-order linear, we did not give much consideration to the involvement of second-order information for consistency. As mentioned in the comments, we recognize that further considering the incorporation of second-order information is a very meaningful direction. If we can find methods to utilize second-order information, we may be able to further refine our definitions of linearity and formula expressions, and estimate the linearity of each module with lower error.
> >
> > Additionally, we might consider iteratively calculating the weight of each module; for example, first calculate the weight of the first layer, then use the fused output features of the first layer to guide the weight calculation of the second layer. This might further exploit the potential connections between modules in different positions. By utilizing second-order information, we might be able to design a new loss function that merges all the weights to be optimized into a simultaneous optimization process. In future work, we will explore more applications of second-order information in our method.
> >
> > We are grateful for this valuable suggestion and will include this discussion in the future work section of the paper.
> >
> > ---
> >
> > **References**
> >
> > [1] Tang, Anke , et al. Merging Multi-Task Models via Weight-Ensembling Mixture of Experts. (2024)
> >
> > [2] Prabhakar, Akshara , et al. LoRA Soups: Merging LoRAs for Practical Skill Composition Tasks. (2024)
> >
> > [3] Sukhbaatar, Sainbayar , et al. Branch-Train-MiX: Mixing Expert LLMs into a Mixture-of-Experts LLM. (2024)
> >
> > ---
> >
> > Please let us know if our response has addressed your concerns or if you have any further questions. We would be grateful for the opportunity to continue discussing with you. Thank you once again for your valuable feedback!

---

> > > ### Comment · Reviewer_SeC4 · 2024-11-21
> > >
> > > Thank you for the response, all my questions have been satisfied.

---

> ### Author Response · Authors · 2024-11-22
>
> Dear Reviewer SeC4,
>
> Thank you again for your time and efforts in reviewing our paper, and for your valuable comments that helped us to strengthen the paper.
>
> Best regards,
>
> Authors of # 3017

---

### Author Response · Authors · 2024-11-26
**General Response About Revision**

First of all, we would like to express our gratitude to all the reviewers for their time and efforts in reviewing our paper. The valuable comments from the reviewers have greatly contributed to strengthening our manuscript.

We have revised our manuscript based on the reviewers' comments and have uploaded it. Here, we summarize our modifications, following the original order presented on this webpage:

1. We have added content related to WEMoE [1] and CAT [2] and included specific comparative tables with further discussions in Appendix A.5.2. (Thanks to Reviewer SeC4)

2. We have provided a more detailed discussion on second-order information in Appendix A.5.3. (Thanks to Reviewer SeC4)

3. We have appropriately modified the introduction to more clearly clarify the linearly properties studied in this paper. Additionally, we have included a comparative table and related discussions in Appendix A.5.1. (Thanks to Reviewers F2K5, g3he, D)

4. We have incorporated descriptions of Git Re-Basin [3], Zipit! [4], and MuDSC [5] and included specific comparative tables with more detailed discussions in Appendix A.5.4. (Thanks to Reviewer F2K5)

5. We have added more discussions regarding the head level and included a more detailed discussion on finely-grained components in Appendix A.5.5. (Thanks to Reviewer F2K5)

6. We have merged Properties 1 (for the model) and Properties 2 (for the module) into more general Properties 1. (Thanks to Reviewer F2K5)

7. We have added the results and discussions of the ablation study on the number of models to be merged to Appendix A.1.3. (Thanks to Reviewers F2K5, g3he)

8. We have included experimental results involving more datasets (including Winogrande, Cosmos QA, and Ropes) and additional models (Qwen-2.5-7B) in Appendix A.1.2. (Thanks to Reviewers F2K5, g3he)

9. We have added experimental results involving more baselines (including TIES [6], Breadcrumbs [7], and Consensus TA [8]) to Appendix A.1.2. (Thanks to Reviewer g3he)

10. We have included the results and discussions of the ablation study on the number of data in Appendix A.1.4. (Thanks to Reviewers g3he, KQV8)

11. We have moved the footnote from the formula to the description below. (Thanks to Reviewer g3he)

12. We have provided a more detailed discussion on non-linear activation functions in Appendix A.5.6. (Thanks to Reviewer KQV8)

13. We have included a detailed discussion on possible methods to improve head-level performance in Appendix A.5.7. (Thanks to Reviewer KQV8)

14. We have added a detailed discussion on the impact of model depth on linearity in Appendix A.5.8. (Thanks to Reviewer KQV8)

15. We have included a detailed discussion on the phenomenon that the non-linearity score for Qwen-2.5-7B is much higher than others in Appendix A.5.9. (Thanks to Reviewer KQV8)


Finally, we would like to thank all the reviewers once again for taking the time to review our paper and providing us with your invaluable feedback. We sincerely appreciate your constructive comments on our work.

Please feel free to ask any remaining questions as we would be glad to provide further clarification. We look forward to any further discussions with all the reviewers.


[1] Tang, Anke , et al. Merging Multi-Task Models via Weight-Ensembling Mixture of Experts. (2024)

[2] Prabhakar, Akshara , et al. LoRA Soups: Merging LoRAs for Practical Skill Composition Tasks. (2024)

[3] Git Re-Basin: Merging Models modulo Permutation Symmetries, ICLR 2023.

[4] Zipit!: Merging Models from Different Tasks without Training. NeurIPS 2023.

[5] Training-Free Pretrained Model Merging, CVPR 2024.

[6] Yadav, Prateek, et al. Ties-merging: Resolving interference when merging models. NIPS 2024.

[7] Davari, MohammadReza, and Eugene Belilovsky. Model breadcrumbs: Scaling multi-task model merging with sparse masks. ECCV 2024.

[8] Wang, Ke, et al. Localizing Task Information for Improved Model Merging and Compression. ICML 2024

---

### Meta-Review · Area_Chair_heEy · 2024-12-24

**Metareview:**

This paper shows that identifying linear submodules can lead to more granular and efficient model merging, and the experimental results demonstrate superiority over existing methods like Task Arithmetic, with valuable ablations on merging granularity. Overall, the reviewers find the presentation is clear, the idea is novel and effective,  and the experimentation and ablation study are of high quality. One concern is that the proposed method could be seen as an extension of prior global linearization work and the improvement is marginal. AC acknowledges the novelty, especially in decomposing models and the novel weight-merging approach. It is thus recommended for acceptance as a poster.

**Additional Comments On Reviewer Discussion:**

Principal concerns put forward by the reviewers were as follows:

1. The finding that submodules possess greater linearity than the overall model might be regarded as self-evident, given that the overall model is assembled from these submodules.

2. The enhancement in performance was rather limited.

3. There was an absence of a comprehensive and profound discussion regarding activation functions.

Following the rebuttal, three reviewers increased their scores to an acceptance level, giving four positive scores of 6. I am inclined to concur with the reviewers. I believe this concept is novel, the method is well-grounded, and the performance improvement is reasonable.

---

### Decision · Program_Chairs · 2025-01-22

Accept (Poster)